# Federated Multi-Objective Learning

**Haibo Yang**
Dept. of Comput. & Info. Sci.
Rochester Institute of Technology
Rochester, NY 14623
hbycis@rit.edu

**Zhuqing Liu**
Dept. of ECE
The Ohio State University
Columbus,OH 43210
liu.9384@osu.edu

**Jia Liu**
Dept. of ECE
The Ohio State University
Columbus,OH 43210
liu@ece.osu.edu

**Chaosheng Dong**
Amazon.com Inc.
Seattle, WA 98109
chaosd@amazon.com

**Michinari Momma**
Amazon.com Inc.
Seattle, WA 98109
michi@amazon.com

## Abstract

In recent years, multi-objective optimization (MOO) emerges as a foundational problem underpinning many multi-agent multi-task learning applications. However, existing algorithms in MOO literature remain limited to centralized learning settings, which do not satisfy the distributed nature and data privacy needs of such multi-agent multi-task learning applications. This motivates us to propose a new federated multi-objective learning (FMOL) framework with multiple clients distributively and collaboratively solving an MOO problem while keeping their training data private. Notably, our FMOL framework allows a different set of objective functions across different clients to support a wide range of applications, which advances and generalizes the MOO formulation to the federated learning paradigm for the first time. For this FMOL framework, we propose two new federated multi-objective optimization (FMOO) algorithms called federated multi-gradient descent averaging (FMGDA) and federated stochastic multi-gradient descent averaging (FSMGDA). Both algorithms allow local updates to significantly reduce communication costs, while achieving the *same* convergence rates as those of their algorithmic counterparts in the single-objective federated learning. Our extensive experiments also corroborate the efficacy of our proposed FMOO algorithms.

## 1 Introduction

In recent years, multi-objective optimization (MOO) has emerged as a foundational problem underpinning many multi-agent multi-task learning applications, such as training neural networks for multiple tasks [1], hydrocarbon production optimization [2], recommendation system [3], tissue engineering [4], and learning-to-rank [5, 6, 7]. MOO aims at optimizing multiple objectives simultaneously, which can be mathematically cast as:

$$\min_{\mathbf{x} \in \mathcal{D}} \mathbf{F}(\mathbf{x}) := [f_1(\mathbf{x}), \cdots, f_S(\mathbf{x})], \tag{1}$$

where $\mathbf{x} \in \mathcal{D} \subseteq \mathbb{R}^d$ is the model parameter, and $f_s : \mathbb{R}^d \to \mathbb{R}$, $s \in [S]$ is one of the objective functions. Compared to conventional single-objective optimization, one key difference in MOO is the coupling and potential conflicts between different objective functions. As a result, there may not exist a common $\mathbf{x}$-solution that minimizes all objective functions. Rather, the goal in MOO is to find a *Pareto stationary solution* that is not improvable for all objectives without sacrificing some objectives. For example, in recommender system designs for e-commerce, the platform needs to consider different

customers with substantially conflicting shopping objectives (price, brand preferences, delivery speed, etc.). Therefore, the platform's best interest is often to find a Pareto-stationary solution, where one cannot deviate to favor one consumer group further without hurting any other group. MOO with conflicting objectives also has natural incarnations in many competitive game-theoretic problems, where the goal is to determine an equilibrium among the conflicting agents in the Pareto sense.

Since its inception dating back to the 1950s, MOO algorithm design has evolved into two major categories: gradient-free and gradient-based methods, with the latter garnering increasing attention in the learning community in recent years due to their better performances (see Section 2 for more detailed discussions). However, despite these advances, all existing algorithms in the current MOO literature remain limited to centralized settings (i.e., training data are aggregated and accessible to a centralized learning algorithm). Somewhat ironically, such centralized settings do *not* satisfy the distributed nature and data privacy needs of many multi-agent multi-task learning applications, which motivates application of MOO in the first place. This gap between the existing MOO approaches and the rapidly growing importance of distributed MOO motivates us to make the first attempt to pursue a new **federated multi-objective learning** (FMOL) framework, with the aim to enable multiple clients to distributively solve MOO problems while keeping their computation and training data private.

So far, however, developing distributed optimization algorithms for FMOL with provable Pareto-stationary convergence remains uncharted territory. There are several key technical challenges that render FMOL far from being a straightforward extension of centralized MOO problems. First of all, due to the distributed nature of FMOL problems, one has to consider and model the *objective heterogeneity* (i.e., different clients could have different sets of objective functions) that is unseen in centralized MOO. Moreover, with local and private datasets being a defining feature in FMOL, the impacts of *data heterogeneity* (i.e., datasets are non-i.i.d. distributed across clients) also need to be mitigated in FMOL algorithm design. Last but not least, under the combined influence of objective and data heterogeneity, FMOL algorithms could be extremely sensitive to small perturbations in the determination of common descent direction among all objectives. This makes the FMOL algorithm design and the associated convergence analysis far more complicated than those of the centralized MOO. Toward this end, a fundamental question naturally arises:

*Under both objective and data heterogeneity in FMOL, is it possible to design effective and efficient algorithms with Pareto-stationary convergence guarantees?*

In this paper, we give an affirmative answer to the above question. Our key contribution is that we propose a new FMOL framework that captures both objective and data heterogeneity, based on which we develop two gradient-based algorithms with provable Pareto-stationary convergence rate guarantees. To our knowledge, our work is the first systematic attempt to bridge the gap between federated learning and MOO. Our main results and contributions are summarized as follows:

- We formalize the first federated multi-objective learning (FMOL) framework that supports both *objective and data heterogeneity* across clients, which significantly advances and generalizes the MOO formulation to the federated learning paradigm. As a result, our FMOL framework becomes a generic model that covers existing MOO models and various applications as special cases (see Section 3.2 for further details). This new FMOL framework lays the foundation to enable us to systematically develop FMOO algorithms with provable Pareto-stationary convergence guarantees.

- For the proposed FMOL framework, we first propose a federated multi-gradient descent averaging (FMGDA) algorithm based on the use of local full gradient evaluation at each client. Our analysis reveals that FMGDA achieves a linear $\mathcal{O}(\exp(-\mu T))$ and a sublinear $\mathcal{O}(1/T)$ Pareto-stationary convergence rates for $\mu$-strongly convex and non-convex settings, respectively. Also, FMGDA employs a two-sided learning rates strategy to significantly lower communication costs (a key concern in the federated learning paradigm). It is worth pointing out that, in the single-machine special case where FMOL degenerates to a centralized MOO problem and FMGDA reduces to the traditional MGD method [8], our results improve the state-of-the-art analysis of MGD by eliminating the restrictive assumptions on the linear search of learning rate and extra sequence convergence. Thus, our results also advance the state of the art in general MOO theory.

- To alleviate the cost of full gradient evaluation in the large dataset regime, we further propose a federated stochastic multi-gradient descent averaging (FSMGDA) algorithm based on the use of stochastic gradient evaluations at each client. We show that FSMGDA achieves $\tilde{\mathcal{O}}(1/T)$ and $\mathcal{O}(1/\sqrt{T})$ Pareto-stationary convergence rate for $\mu$-strongly convex and non-convex settings, respectively. We establish our convergence proof by proposing a new $(\alpha, \beta)$-Lipschitz continuous

Table 1: Convergence rate results (shaded parts are our results) comparisons.

| Methods | Strongly Convex | | Non-convex | |
|---|---|---|---|---|
| | Rate | Assumption$^*$ | Rate | Assumption$^*$ |
| MGD [8] | $\mathcal{O}(r^T)$ $^\#$ | Linear search & sequence convergence | $\mathcal{O}(1/T)$ | Linear search & sequence convergence |
| SMGD [9] | $\mathcal{O}(1/T)$ | First moment bound & Lipschitz continuity of $\lambda$ | Not provided | Not provided |
| FMGDA | $\mathcal{O}(\exp(-\mu T))$ $^\#$ | Not needed | $\mathcal{O}(1/T)$ | Not needed |
| FSMGDA | $\tilde{\mathcal{O}}(1/T)$ | $(\alpha, \beta)$-Lipschitz continuous stochastic gradient | $\mathcal{O}(1/\sqrt{T})$ | $(\alpha, \beta)$-Lipschitz continuous stochastic gradient |

$^\#$Notes on constants: $\mu$ is the strong convexity modulus; $r$ is a constant depends on $\mu$, s.t., $r \in (0, 1)$.
$^*$Assumption short-hands: "Linear search": learning rate linear search [8]; "Sequence convergence": $\{\mathbf{x}_t\}$ converges to $\mathbf{x}^*$ [8]; "First moment bound" (Asm. 5.2(b) [9]): $\mathbb{E}[\|\nabla f(\mathbf{x}, \xi) - \nabla f(\mathbf{x})\|] \leq \eta(a + b\|\nabla f(\mathbf{x})\|)$;"Lipschitz continuity of $\lambda$" (Asm. 5.4 [9]): $\|\boldsymbol{\lambda}_k - \boldsymbol{\lambda}_t\| \leq \beta \|[(\nabla f_1(\mathbf{x}_k) - \nabla f_1(\mathbf{x}_t))^T, \ldots, (\nabla f_S(\mathbf{x}_k) - \nabla f_S(\mathbf{x}_t))^T]\|$; "$(\alpha, \beta)$-Lipschitz continuous stochastic gradient": see Asm. 4.

stochastic gradient assumption (cf. Assumption 4), which relaxes the strong assumptions on first moment bound and Lipschitz continuity on common descent directions in [9]. We note that this new $(\alpha, \beta)$-Lipschitz continuous stochastic gradient assumption can be viewed as a natural extension of the classical Lipschitz-continuous gradient assumption and could be of independent interest.

The rest of the paper is organized as follows. In Section 2, we review related works. In Section 3, we introduce our FMOL framework and two gradient-based algorithms (FMGDA and FSMGDA), which are followed by their convergence analyses in Section 4. We present the numerical results in Section 5 and conclude the work in Section 6. Due to space limitations, we relegate all proofs and some experiments to supplementary material.

## 2 Related work

In this section, we will provide an overview on algorithm designs for MOO and federated learning (FL), thereby placing our work in a comparative perspective to highlight our contributions and novelty.

**1) Multi-objective Optimization (MOO):** As mentioned in Section 1, since federated/distributed MOO has not been studied in the literature, all existing works we review below are centralized MOO algorithms. Roughly speaking, MOO algorithms can be grouped into two main categories. The first line of works are gradient-free methods (e.g., evolutionary MOO algorithms and Bayesian MOO algorithms [10, 11, 12, 13]). These methods are more suitable for small-scale problems but less practical for high-dimensional MOO models (e.g., deep neural networks). The second line of works focus on gradient-based approaches [14, 15, 8, 16, 9, 17], which are more practical for high-dimensional MOO problems. However, while having received increasing attention from the community in recent years, Pareto-stationary convergence analysis of these gradient-based MOO methods remains in its infancy.

Existing gradient-based MOO methods can be further categorized as i) multi-gradient descent (MGD) algorithms with full gradients and ii) stochastic multi-gradient descent (SMGD) algorithms. It has been shown in [8] that MGD methods achieve $\mathcal{O}(r^T)$ for some $r \in (0, 1)$ and $\mathcal{O}(1/T)$ Pareto-stationary convergence rates for $\mu$-strongly convex and non-convex functions, respectively. However, these results are established under the unconventional linear search of learning rate and sequence convergence assumptions, which are difficult to verify in practice. In comparison, FMGDA achieves a linear rate without needing such assumptions. For SMGD methods, the Pareto-stationary convergence analysis is further complicated by the stochastic gradient noise. Toward this end, an $\mathcal{O}(1/T)$ rate analysis for SMGD was provided in [9] based on rather strong assumptions on a first-moment bound and Lipschtiz continuity of common descent direction. As a negative result, it was shown in [9] and [18] that the common descent direction needed in the SMGD method is likely to be a biased estimation, which may cause divergence issues.

In contrast, our FSMGDA achieves state-of-the-art $\tilde{\mathcal{O}}(1/T)$ and $\mathcal{O}(1/\sqrt{T})$ convergence rates for strongly-convex and non-convex settings, respectively, under a much milder assumption on Lipschtiz continuous stochastic gradients. For easy comparisons, we summarize our results and the existing works in Table 1. It is worth noting recent works [18, 19, 20] established faster convergence rates in

the centralized MOO setting by using acceleration techniques, such as momentum, regularization and bi-level formulation. However, due to different settings and focuses, these results are orthogonal to ours and thus not directly comparable. Also, since acceleration itself is a non-trivial topic and could be quite brittle if not done right, in this paper, we focus on the basic and more robust stochastic gradient approach in FMOL. But for a comprehensive comparison on assumptions and main results of accelerated centralized MOO, we refer readers to Appendix A for further details.

**Federated Learning (FL):** Since the seminal work by [21], FL has emerged as a popular distributed learning paradigm. Traditional FL aims at solving single-objective minimization problems with a large number of clients with decentralized data. Recent FL algorithms enjoy both high communication efficiency and good generalization performance [21, 22, 23, 24, 25, 26]. Theoretically, many FL methods have the same convergence rates as their centralized counterparts under different FL settings [27, 28, 29, 30]. Recent works have also considered FL problems with more sophisticated problem structures, such as min-max learning [31, 32], reinforcement learning [33], multi-armed bandits [34], and bilevel and compositional optimization [35]. Although not directly related, classic FL has been reformulated in the form of MOO[36], which allows the use of a MGD-type algorithm instead of vanilla local SGD to solve the standard FL problem. We will show later that this MOO reformulation is a special case of our FMOL framework. So far, despite a wide range of applications (see Section 3.2 for examples), there remains a lack of a general FL framework for MOO. This motivates us to bridge the gap by proposing a general FMOL framework and designing gradient-based methods with provable Pareto-stationary convergence rates.

# 3 Federated multi-objective learning

## 3.1 Multi-objective optimization: A primer

As mentioned in Section 1, due to potential conflicts among the objective functions in MOO problem in (1), MOO problems adopt the the notion of Pareto optimality:

**Definition 1** ((Weak) Pareto Optimality). *For any two solutions* $\mathbf{x}$ *and* $\mathbf{y}$*, we say* $\mathbf{x}$ *dominates* $\mathbf{y}$ *if and only if* $f_s(\mathbf{x}) \leq f_s(\mathbf{y}), \forall s \in [S]$ *and* $f_s(\mathbf{x}) < f_s(\mathbf{y}), \exists s \in [S]$. *A solution* $\mathbf{x}$ *is Pareto optimal if it is not dominated by any other solution. One solution* $\mathbf{x}$ *is weakly Pareto optimal if there does not exist a solution* $\mathbf{y}$ *such that* $f_s(\mathbf{x}) > f_s(\mathbf{y}), \forall s \in [S]$.

Similar to solving single-objective non-convex optimization problems, finding a Pareto-optimal solution in MOO is NP-Hard in general. As a result, it is often of practical interest to find a solution satisfying Pareto-stationarity (a necessary condition for Pareto optimality) stated as follows [14, 37]:

**Definition 2** (Pareto Stationarity). *A solution* $\mathbf{x}$ *is said to be Pareto stationary if there is no common descent direction* $\mathbf{d} \in \mathbb{R}^d$ *such that* $\nabla f_s(\mathbf{x})^\top \mathbf{d} < 0, \forall s \in [S]$.

Note that for strongly convex functions, Pareto stationary solutions are also Pareto optimal. Following Definition 2, gradient-based MOO algorithms typically search for a common descent direction $\mathbf{d} \in \mathbb{R}^d$ such that $\nabla f_s(\mathbf{x})^\top \mathbf{d} \leq 0, \forall s \in [S]$. If no such a common descent direction exists at $\mathbf{x}$, then $\mathbf{x}$ is a Pareto stationary solution. For example, MGD [15] searches for an optimal weight $\boldsymbol{\lambda}^*$ of gradients $\nabla \mathbf{F}(\mathbf{x}) \triangleq \{\nabla f_s(\mathbf{x}), \forall s \in [S]\}$ by solving $\boldsymbol{\lambda}^*(\mathbf{x}) = \operatorname{argmin}_{\boldsymbol{\lambda} \in C} \|\boldsymbol{\lambda}^\top \nabla \mathbf{F}(\mathbf{x})\|^2$. Then, a common descent direction can be chosen as: $\mathbf{d} = \boldsymbol{\lambda}^\top \nabla \mathbf{F}(\mathbf{x})$. MGD performs the iterative update rule: $\mathbf{x} \leftarrow \mathbf{x} - \eta \mathbf{d}$ until a Pareto stationary point is reached, where $\eta$ is a learning rate. SMGD [9] also follows the same process except for replacing full gradients by stochastic gradients. For MGD and SMGD methods, it is shown in [8] and [18] show that if $\|\boldsymbol{\lambda}^\top \nabla \mathbf{F}(\mathbf{x})\| = 0$ for some $\boldsymbol{\lambda} \in C$, where $C \triangleq \{\mathbf{y} \in [0,1]^S, \sum_{s \in [S]} y_s = 1\}$, then $\mathbf{x}$ is a Pareto stationary solution. Thus, $\|\mathbf{d}\|^2 = \|\boldsymbol{\lambda}^\top \nabla \mathbf{F}(\mathbf{x})\|^2$ can be used as a metric to measure the convergence of non-convex MOO algorithms [8, 18, 19]. On the other hand, for more tractable strongly convex MOO problems, the optimality gap $\sum_{s \in [S]} \lambda_s [f_s(\mathbf{x}) - f_s(\mathbf{x}^*)]$ is typically used as the metric to measure the convergence of an algorithm [9], where $\mathbf{x}^*$ denotes the Pareto optimal point. We summarize and compare different convergence metrics as well as assumptions in MOO, detailed in Appendix A.

## 3.2 A general federated multi-objective learning framework

With the MOO preliminaries in Section 3.1, we now formalize our general federated multi-objective learning (FMOL) framework. For a system with $M$ clients and $S$ tasks (objectives), our FMOL framework can be written as:

$$\min_{\mathbf{x}} \quad \mathrm{Diag}(\mathbf{F}\mathbf{A}^{\top}), \tag{2}$$

$$\mathbf{F} \triangleq \begin{bmatrix} f_{1,1} & \cdots & f_{1,M} \\ \vdots & \ddots & \vdots \\ f_{S,1} & \cdots & f_{S,M} \end{bmatrix}_{S \times M}, \mathbf{A} \triangleq \begin{bmatrix} a_{1,1} & \cdots & a_{1,M} \\ \vdots & \ddots & \vdots \\ a_{S,1} & \cdots & a_{S,M} \end{bmatrix}_{S \times M},$$

where matrix $\mathbf{F}$ groups all potential objectives $f_{s,i}(\mathbf{x})$ for each task $s$ at each client $i$, and $\mathbf{A} \in \{0,1\}^{S \times M}$ is a *binary* objective indicator matrix, with each element $a_{s,i} = 1$ if task $s$ is of client $i$'s interest and $a_{s,i} = 0$ otherwise. For each task $s \in [S]$, the global objective function $f_s(\mathbf{x})$ is the average of local objectives over all related clients, i.e., $f_s(\mathbf{x}) \triangleq \frac{1}{|R_s|} \sum_{i \in R_s} f_{s,i}(\mathbf{x})$, where $R_s = \{i : a_{s,i} = 1, i \in [M]\}$. Note that, for notation simplicity, here we use simple average in $f_s(\mathbf{x})$, which corresponds to the balanced dataset setting. Our FMLO framework can be directly extended to imbalanced dataset settings by using weighted average proportional to dataset sizes of related clients. For a client $i \in [M]$, its objectives of interest are $\{f_{s,i}(\mathbf{x}) : a_{s,i} = 1, s \in [S]\}$, which is a subset of $[S]$.

We note that FMOL generalizes MOO to the FL paradigm, which includes many existing MOO problems as special cases and corresponds to a wide range of applications.

- If each client has only one distinct objective, i.e., $\mathbf{A} = \mathbb{I}_M$, $S = M$, then $\mathrm{Diag}(\mathbf{F}\mathbf{A}^{\top}) = [f_1(\mathbf{x}), \ldots, f_S(\mathbf{x})]^{\top}$, where each objective $f_s(\mathbf{x}), s \in [S]$ is optimized only by client $s$. This special FMOL setting corresponds to the conventional multi-task learning and federated learning. Indeed, [1] and [38] formulated a multi-task learning problem as MOO and considered Pareto optimal solutions with various trade-offs. [36] also formulated FL as as distributed MOO problems. Other examples of this setting include bi-objective formulation of offline reinforcement learning [39] and decentralized MOO [40].

- If all clients share the same $S$ objectives, i.e., $\mathbf{A}$ is an all-one matrix, then $\mathrm{Diag}(\mathbf{F}\mathbf{A}^{\top}) = \left[\frac{1}{M} \sum_{i \in [M]} f_{1,i}(\mathbf{x}), \ldots, \frac{1}{M} \sum_{i \in [M]} f_{S,i}(\mathbf{x})\right]^{\top}$. In this case, FMOL reduces to federated MOO problems with decentralized data that jointly optimizing fairness, privacy, and accuracy [41, 42, 43], as well as MOO with decentralized data under privacy constraints (e.g., machine reassignment among data centres [44] and engineering problems [45, 46, 47, 48]).

- If each client has a different subset of objectives (i.e., objective heterogeneity), FMLO allows distinct preferences at each client. For example, each customer group on a recommender system in e-commerce platforms might have different combinations of shopping preferences, such as product price, brand, delivery speed, etc.

## 3.3 Federated Multi-Objective Learning Algorithms

Upon formalizing our FMOL framework, our next goal is to develop gradient-based algorithms for solving large-scale high-dimensional FMOL problems with *provable* Pareto stationary convergence guarantees and low communication costs. To this end, we propose two FMOL algorithms, namely federated multiple gradient descent averaging (FMGDA) and federated stochastic multiple gradient descent averaging (FSMGDA) as shown in Algorithm 1. We summarize our key notation in Table 3 in Appendix to allow easy references for readers.

As shown in Algorithm 1, in each communication round $t \in [T]$, each client synchronizes its local model with the current global model $\mathbf{x}_t$ from the server (cf. Step 1). Then each client runs $K$ local steps based on local data for all effective objectives (cf. Step 2) with two options: i) for FMGDA, each local step performs local full gradient descent, i.e., $\mathbf{x}_{s,i}^{t,k+1} = \mathbf{x}_{s,i}^{t,k} - \eta_L \nabla f_{s,i}(\mathbf{x}_{s,i}^{t,k}), \forall s \in S_i$; ii) For FSMGDA, the local step performs stochastic gradient descent, i.e., $\mathbf{x}_{s,i}^{t,k+1} = \mathbf{x}_{s,i}^{t,k} - \eta_L \nabla f_{s,i}(\mathbf{x}_{s,i}^{t,k}, \xi_i^{t,k}), \forall s \in S_i$, where $\xi_i^{t,k}$ denotes a random sample in local step $k$ and round $t$ at client $i$. Upon finishing $K$ local updates, each client returns the accumulated update $\Delta_{s,i}^{t}$ for each effective objective to the server (cf. Step 3). Then, the server aggregates all returned $\Delta$-updates from

---

**Algorithm 1** Federated (Stochastic) Multiple Gradient Descent Averaging (FMGDA/FSMGDA).

---

**At Each Client $i$:**

1. Synchronize local models $\mathbf{x}_{s,i}^{t,0} = \mathbf{x}_t, \forall s \in S_i$.

2. Local updates: for all $s \in S_i$, for $k = 1, \ldots, K$,
    (FMGDA): $\mathbf{x}_{s,i}^{t,k} = \mathbf{x}_{s,i}^{t,k-1} - \eta_L \nabla f_{s,i}(\mathbf{x}_{s,i}^{t,k-1})$.
    (FSMGDA): $\mathbf{x}_{s,i}^{t,k} = \mathbf{x}_{s,i}^{t,k-1} - \eta_L \nabla f_{s,i}(\mathbf{x}_{s,i}^{t,k-1}, \xi_i^{t,k})$.

3. Return accumulated updates to server $\{\Delta_{s,i}^t, s \in S_i\}$:
    (FMGDA): $\Delta_{s,i}^t = \sum_{k \in [K]} \nabla f_{s,i}(\mathbf{x}_{s,i}^{t,k})$.
    (FSMGDA): $\Delta_{s,i}^t = \sum_{k \in [K]} \nabla f_{s,i}(\mathbf{x}_{s,i}^{t,k}, \xi_i^{t,k})$.

**At the Server:**

4. Receive accumulated updates $\{\Delta_{s,i}^t, \forall s \in S_i, \forall i \in [M]\}$.

5. Compute $\Delta_s^t = \frac{1}{|R_s|} \sum_{i \in R_s} \Delta_{s,i}^t, \forall s \in [S]$, where $R_s = \{i : a_{s,i} = 1, i \in [M]\}$.

6. Compute $\boldsymbol{\lambda}_t^* \in [0,1]^S$ by solving

$$\min_{\boldsymbol{\lambda}_t \geq \mathbf{0}} \left\| \sum_{s \in [S]} \lambda_s^t \Delta_s^t \right\|^2, \quad \text{s.t.} \sum_{s \in [S]} \lambda_s^t = 1. \tag{3}$$

7. Let $\mathbf{d}_t = \sum_{s \in [S]} \lambda_s^{t,*} \Delta_s^t$ and update the global model as: $\mathbf{x}_{t+1} = \mathbf{x}_t - \eta_t \mathbf{d}_t$, with a global learning rate $\eta_t$.

---

the clients to obtain the overall updates $\Delta_s^t$ for each objective $s \in [S]$ (cf. Steps 4 and 5), which will be used in solving a convex quadratic optimization problem with linear constraints to obtain an approximate common descent direction $\mathbf{d}_t$ (cf. Step 6). Lastly, the global model is updated following the direction $\mathbf{d}_t$ with global learning rate $\eta_t$ (cf. Step 7).

Two remarks on Algorithm 1 are in order. First, we note that a two-sided learning rates strategy is used in Algorithm 1, which decouples the update schedules of local and global model parameters at clients and server, respectively. As shown in Section 4 later, this two-sided learning rates strategy enables better convergence rates by choosing appropriate learning rates. Second, to achieve low communication costs, Algorithm 1 leverages $K$ local updates at each client and infrequent periodic communications between each client and the server. By adjusting the two-sided learning rates appropriately, the $K$-value can be made large to further reduce communication costs.

## 4 Pareto stationary convergence analysis

In this section, we analyze the Pareto stationary convergence performance for our FMGDA and FSMGDA algorithms in Sections 4.1 and 4.2, respectively, each of which include non-convex and strongly convex settings.

### 4.1 Pareto stationary convergence of FMGDA

In what follows, we show FMGDA enjoys linear rate $\tilde{\mathcal{O}}(\exp(-\mu T))$ for $\mu$-strongly convex functions and sub-linear rate $\mathcal{O}(\frac{1}{T})$ for non-convex functions.

**1) FMGDA: The Non-convex Setting.** Before presenting our Pareto stationary convergence results for FMGDA, we first state serveral assumptions as follows:

**Assumption 1.** *(L-Lipschitz continuous) There exists a constant $L > 0$ such that $\|\nabla f_s(\mathbf{x}) - \nabla f_s(\mathbf{y})\| \leq L\|\mathbf{x} - \mathbf{y}\|, \forall \mathbf{x}, \mathbf{y} \in \mathbb{R}^d, s \in [S]$.*

**Assumption 2.** *(Bounded Gradient) The gradient of each objective at any client is bounded, i.e., there exists a constant $G > 0$ such that $\|\nabla f_{s,i}(\mathbf{x})\|^2 \leq G^2, \forall s \in [S], i \in [M]$.*

With the assumptions above, we state the Pareto stationary convergence of FMGDA for non-convex FMOL as follows:

**Theorem 1** (FMGDA for Non-convex FMOL)**.** *Let $\eta_t = \eta \leq \frac{3}{2(1+L)}$. Under Assumptions 1 and 2, if at least one function $f_s, s \in [S]$ is bounded from below by $f_s^{\min}$, then the sequence $\{\mathbf{x}_t\}$ output by FMGDA satisfies:* $\min_{t \in [T]} \|\bar{\mathbf{d}}_t\|^2 \leq \frac{16(f_s^0 - f_s^{\min})}{T\eta} + \delta$, *where* $\delta \triangleq \frac{16\eta_L^2 K^2 L^2 G^2 (1+S^2)}{\eta}$.

In non-convex functions, we use $\|\bar{\mathbf{d}}_t\|^2$ as the metrics for FMOO, where $\bar{\mathbf{d}}_t = \boldsymbol{\lambda}_t^T \nabla(\text{Diag}(\mathbf{FA}^\top))$ and $\boldsymbol{\lambda}_t$ is calculated by the quadratic programming problem 3 based on accumulated (stochastic) gradients $\Delta_t$. We compare different metrics for MOO in Appendix A. The convergence bound in Theorem 1 contains two parts. The first part is an optimization error, which depends on the initial point and vanishes as $T$ increases. The second part is due to local update steps $K$ and data heterogeneity $G$, which can be mitigated by carefully choosing the local learning rate $\eta_L$. Specifically, the following Pareto stationary convergence rate of FMGDA follows immediately from Theorem 1 with an appropriate choice of local learning rate $\eta_L$:

**Corollary 2.** *With a constant global learning rate $\eta_t = \eta, \forall t$, and a local learning rate $\eta_L = \mathcal{O}(1/\sqrt{T})$, the Pareto stationary convergence rate of FMGDA is $(1/T) \sum_{t \in [T]} \|\bar{\mathbf{d}}_t\|^2 = \mathcal{O}(1/T)$.*

Several interesting insights of Theorem 1 and Corollary 2 are worth pointing out: **1)** We note that FMGDA achieves a Pareto stationary convergence rate $\mathcal{O}(1/T)$ for non-convex FMOL, which is the *same* as the Pareto stationary rate of MGD for centralized MOO and the *same* convergence rate of gradient descent (GD) for single objective problems. This is somewhat surprising because FMGDA needs to handle more complex objective and data heterogeneity under FMOL; **2)** The two-sided learning rates strategy decouples the operation of clients and server by utilizing different learning rate schedules, thus better controlling the errors from local updates due to data heterogeneity; **3)** Note that in the single-client special case, FMGDA degenerates to the basic MGD algorithm. Hence, Theorem 1 directly implies a Pareto stationary convergence bound for MGD by setting $\delta = 0$ due to no local updates in centralized MOO. This convergence rate bound is consistent with that in [8]. However, we note that our result is achieved *without* using the linear search step for learning rate [8], which is much easier to implement in practice (especially for deep learning models); **4)** Our proof is based on standard assumptions in first-order optimization, while previous works require strong and unconventional assumptions. For example, a convergence of $\mathbf{x}$-sequence is assumed in [8].

**2) FMGDA: The Strongly Convex Setting.** Now, we consider the strongly convex setting for FMOL, which is more tractable but still of interest in many learning problems in practice. In the strongly convex setting, we have the following additional assumption:

**Assumption 3.** *($\mu$-Strongly Convex Function) Each objective $f_s(\mathbf{x}), s \in [S]$ is a $\mu$-strongly convex function, i.e., $f_s(\mathbf{y}) \geq f_s(\mathbf{x}) + \nabla f_s(\mathbf{x})(\mathbf{y} - \mathbf{x}) + \frac{\mu}{2}\|\mathbf{y} - \mathbf{x}\|^2$ for some $\mu > 0$.*

For more tractable strongly-convex FMOL problems, we show that FMGDA achieves a stronger Pareto stationary convergence performance as follows:

**Theorem 3** (FMGDA for $\mu$-Strongly Convex FMOL)**.** *Let $\eta_t = \eta$ such that $\eta \leq \frac{3}{2(1+L)}, \eta \leq \frac{1}{2L+\mu}$ and $\eta \geq \frac{1}{\mu T}$. Under Assumptions 1- 3, pick $\mathbf{x}_t$ as the final output of the FMGDA algorithm with weights $w_t = (1 - \frac{\mu\eta}{2})^{1-t}$. Then, it holds that $\mathbb{E}[\Delta_Q^t] \leq \|\mathbf{x}_0 - \mathbf{x}_*\|^2 \mu \exp(-\frac{\eta\mu T}{2}) + \delta$, where $\Delta_Q^t \triangleq \sum_{s \in [S]} \lambda_s^{t,*} [f_s(\mathbf{x}_t) - f_s(\mathbf{x}_*)]$ and $\delta = \frac{8\eta_L^2 K^2 L^2 G^2 S^2}{\mu} + 2\eta_L^2 K^2 L^2 G^2$.*

Theorem 3 immediately implies following Pareto stationary convergence rate for FMGDA with a proper choice of local learning rate:

**Corollary 4.** *If $\eta_L$ is chosen sufficiently small such that $\delta = \mathcal{O}(\mu \exp(-\mu T))$, then the Pareto stationary convergence rate of FMGDA is $\mathbb{E}[\Delta_Q^t] = \mathcal{O}(\mu \exp(-\mu T))$.*

Again, several interesting insights can be drawn from Theorem 3 and Corollary 4. First, for strongly convex FMOL, FMGDA achieves a linear convergence rate $\mathcal{O}(\mu \exp(-\mu T))$, which again matches those of MGD for centralized MOO and GD for single-objective problems. Second, compared with the non-convex case, the convergence bounds suggest FMGDA could use a larger local learning rate for non-convex functions thanks to our two-sided learning rates design. A novel feature of FMGDA for strongly convex FMOL is the randomly chosen output $x_t$ with weight $w_t$ from the $\mathbf{x}_t$-trajectory, which is inspired by the classical work in stochastic gradient descent (SGD) [49]. Note that, for implementation in practice, one does not need to store all $\mathbf{x}_t$-values. Instead, the algorithm can be implemented by using a random clock for stopping [49].

## 4.2 Pareto stationary convergence of FSMGDA

While enjoying strong performances, FMGDA uses local full gradients at each client, which could be costly in the large dataset regime. Thus, it is of theoretical and practical importance to consider the stochastic version of FMGDA, i.e., federated stochastic multi-gradient descent averaging (FSMGDA).

**1) FSMGDA: The Non-convex Setting.** A fundamental challenge in analyzing the Pareto stationarity convergence of FSMGDA and other stochastic multi-gradient descent (SMGD) methods stems from bounding the error of the common descent direction estimation, which is affected by both $\boldsymbol{\lambda}_t^*$ (obtained by solving a quadratic programming problem) and the stochastic gradient variance. In fact, it is shown in [9] and [18] that the stochastic common descent direction in SMGD-type methods could be biased, leading to divergence issues. To address these challenges, in this paper, we propose to use a *new* assumption on the stochastic gradients, which is stated as follows:

**Assumption 4** (($\alpha, \beta$)-Lipschitz Continuous Stochastic Gradient). *A function $f$ has ($\alpha, \beta$)-Lipschitz continuous stochastic gradients if there exist two constants $\alpha, \beta > 0$ such that, for any two independent training samples $\xi$ and $\xi'$, $\mathbb{E}[\|\nabla f(\mathbf{x}, \xi) - \nabla f(\mathbf{y}, \xi')\|^2] \leq \alpha \|\mathbf{x} - \mathbf{y}\|^2 + \beta\sigma^2$.*

In plain language, Assumption 4 says that the stochastic gradient estimation of an objective does not change too rapidly. We note that the ($\alpha, \beta$)-Lipschitz continuous stochastic gradient assumption is a natural extension of the classic $L$-Lipschitz continuous gradient assumption (cf. Assumption 1) and generalizes several assumptions of SMGD convergence analysis in previous works. We note that Assumption 1 is not necessarily too hard to satisfy in practice. For example, when the underlying distribution of training samples $\xi$ has a bounded support (typically a safe assumption for most applications in practice due to the finite representation limit of computing systems), suppose that Assumption 1 holds (also a common assumption in the optimization literature), then for any given $\mathbf{x}$ and $\mathbf{y}$, the left-hand-side of the inequality in Assumption 4 is bounded due to the L-smoothness in Assumption 1. In this case, there always exist a sufficiently large $\alpha$ and a $\beta$ such that the right-hand-side of the inequality in Assumption 1 holds. Please see Appendix A for further details. In addition, we need the following assumptions for the stochastic gradients, which are commonly used in standard SGD-based analyses [49, 50, 51, 52].

**Assumption 5.** *(Unbiased Stochastic Estimation) The stochastic gradient estimation is unbiased for each objective among clients, i.e., $\mathbb{E}[\nabla f_{s,i}(\mathbf{x}, \xi)] = \nabla f_{s,i}(\mathbf{x}), \forall s \in [S], i \in [M]$.*

**Assumption 6.** *(Bounded Stochastic Gradient) The stochastic gradients satisfiy $\mathbb{E}[\|\nabla f_{s,i}(\mathbf{x}, \xi)\|^2] \leq D^2, \forall s \in [S], i \in [M]$ for some constant $D > 0$.*

With the assumptions above, we now state the Pareto stationarity convergence of FSMGDA as follows:

**Theorem 5** (FSMGDA for Non-convex FMOL). *Let $\eta_t = \eta \leq \frac{3}{2(1+L)}$. Under Assumptions 4–6, if an objective $f_s$ is bounded from below by $f_s^{\min}$, then the sequence $\{\mathbf{x}_t\}$ output by FSMGDA satisfies:* $\min_{t \in [T]} \mathbb{E}\left\|\bar{\mathbf{d}}_t\right\|^2 \leq \frac{8(f_s^0 - f_s^{\min})}{\eta T} + \delta$, *where $\delta = (2S^2 + 4)(\alpha \eta_L^2 K^2 D^2 + \beta\sigma^2)$.*

Theorem 5 immediately implies an $\mathcal{O}(1/\sqrt{T})$ convergence rate of FSMGDA for non-convex FMOL:

**Corollary 6.** *With a constant global learning rate $\eta_t = \eta = \mathcal{O}(1/\sqrt{T})$, $\forall t$ and a local learning rate $\eta_L = \mathcal{O}\left(1/T^{1/4}\right)$, and if $\beta = \mathcal{O}(\eta)$, the Pareto stationarity convergence rate of FSMGDA is $\min_{t \in [T]} \mathbb{E}\|\bar{\mathbf{d}}_t\|^2 = \mathcal{O}(1/\sqrt{T})$.*

**2) The Strongly Convex Setting:** For more tractable strongly convex FMOL problems, we can show that FSMGDA achieve stronger convergence results as follows:

**Theorem 7** (FSMGDA for $\mu$-Strongly Convex FMOL). *Let $\eta_t = \eta = \Omega(\frac{1}{\mu T})$. Under Assumptions 3, 5 and 6, pick $\mathbf{x}_t$ as the final output of the FSMGDA algorithm with weight $w_t = (1 - \frac{\mu\eta}{2})^{1-t}$. Then, it holds that: $\mathbb{E}[\Delta_Q^t] \leq \|\mathbf{x}_0 - \mathbf{x}_*\|^2 \mu \exp(-\frac{\eta}{2}\mu T) + \delta$, where $\Delta_Q^t = \sum_{s \in [S]} \lambda_s^{t,*} [f_s(\mathbf{x}_t) - f_s(\mathbf{x}_*)]$ and $\delta = \frac{1}{\mu} S^2 (\alpha \eta_L^2 K^2 D^2 + \beta\sigma^2) + \frac{\eta S^2 D^2}{2}$.*

The following Pareto station convergence rate of FSMGDA follows immediately from Theorem 7:

**Corollary 8.** *Choose $\eta_L = \mathcal{O}(\frac{1}{\sqrt{T}})$ and $\eta = \Theta(\frac{\log(\max(1, \mu^2 T))}{\mu T})$. If $\beta = \mathcal{O}(\eta)$, then the Pareto stationary convergence rate of FSMGDA is $\mathbb{E}[\Delta_Q^t] \leq \tilde{\mathcal{O}}(1/T)$.*

Corollary 8 says that, With proper learning rates, FSMGDA achieves $\tilde{\mathcal{O}}(1/T)$ Pareto stationary convergence rate (i.e., ignoring logarithmic factors) for strongly convex FMOL. Also, in the single-client special case with no local updates, FSMGDA reduces to the SMGD algorithm and $\delta = \frac{4}{\mu}\beta S^2 \sigma^2 + \frac{\eta S^2 D^2}{2}$ in this case. Then, Theorem 7 implies an $\tilde{\mathcal{O}}(\frac{1}{T})$ Pateto stationarity convergence rate for SMGD for strongly convex MOO problems, which is consistent with previous works [9]. However, our convergence rate proof uses a more conventional $(\alpha, \beta)$-Lipschitz stochastic gradient assumption, rather than the unconventional assumptions on the first moment bound and Lipschitz continuity of common descent directions in [9].

## 5  Numerical results

In this section, we show the main numerical experiments of our FMGDA and FSMGDA algorithms in different datasets, while relegating the experimental settings and details to the appendix.

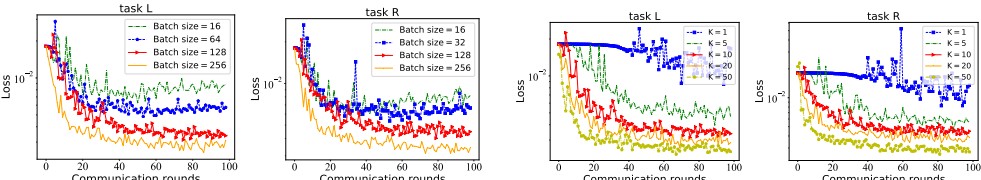

(a) Training loss convergence in terms of communication rounds with different batch-sizes under non-i.i.d. data partition in MultiMNIST.

(b) The impacts of local update number $K$ on training loss convergence in terms of communication rounds.

Figure 1: Training loss convergence comparison.

**1) Ablation Experiments on Two-Tasks FMOL:** *1-a) Impacts of Batch Size on Convergence:* First, we compare the convergence results in terms of the number of communication rounds using the "MultiMNIST" dataset [53] with two tasks (L and R) as objectives. We test our algorithms with four different cases with batch sizes being $[16, 64, 128, 256]$. To reduce computational costs in this experiment, the dataset size for each client is limited to $256$. Hence, the batch size $256$ corresponds to FMGDA and all other batch sizes correspond to FSMGDA. As shown in Fig. 1(a), under non-i.i.d. data partition, both FMGDA and FSMGDA algorithms converge. Also, the convergence speed of the FSMGDA algorithm increases as the batch size gets larger. These results are consistent with our theoretical analyses as outlined in Theorems 1 and 5.

*1-b) Impacts of Local Update Steps on Convergence:* Next, we evaluate our algorithms with different numbers of local update steps $K$. As shown in Fig. 1(b) and Table 2, both algorithms converge faster as the number of the local steps $K$ increases. This is because both algorithms effectively run more iterative updates as $K$ gets large.

*1-c) Comparisons between FMOL and Centralized MOO:* Since this work is the first that investigates FMOL, it is also interesting to empirically compare the differences between FMOL and centralized MOO methods. In Fig. 2(a), we compare the training loss of FMGDA and FSMGDA with those of the centralized MGD and SMGD methods after 100 communication rounds. For fair comparisons, the centralized MGD and SMGD methods use $\sum_i^M |S_i|$ batch-sizes and run $K \times T$ iterations. Our results indicate that FMGDA and MGD produce similar results, while the performance of FSMGDA is slightly worse than that of SMGD due to FSMGDA's sensitivity to objective and data heterogeneity in stochastic settings. These numerical results confirm our theoretical convergence analysis.

Table 2: Communication rounds needed for $10^{-2}$ loss.

|          | i.i.d.      |             | non-i.i.d.  |             |
| -------- | ----------- | ----------- | ----------- | ----------- |
|          | Task L      | Task R      | Task L      | Task R      |
| $K = 1$  | 82          | 84          | 96          | 82          |
| $K = 5$  | 18(4.6×)    | 20(4.2×)    | 24(4.0×)    | 20(4.1×)    |
| $K = 10$ | 10(8.2×)    | 9(9.3×)     | 13(7.4×)    | 10(8.2×)    |
| $K = 20$ | 5(16.4×)    | 5(16.8×)    | 6(16.0×)    | 5(16.4×)    |

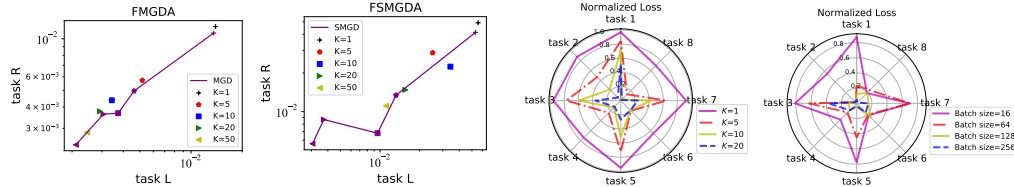

(a) 100 communication rounds with various local steps $K$, corresponding federated and centralized settings share the same marker shape.

(b) Normalized loss with the River Flow datasets.

Figure 2: Training losses comparison

**2) Experiments on Larger FMOL:** We further test our algorithms on FMOL problems of larger sizes. In this experiment, we use the River Flow dataset[54], which contains *eight* tasks in this problem. To better visualize 8 different tasks, we illustrate the normalized loss in radar charts in Fig. 2(b). In this 8-task setting, we can again verify that more local steps $K$ and a larger training batch size lead to faster convergence. In the appendix, we also verify the effectiveness of our FMGDA and FSMGDA algorithms in CelebA [55] (*40 tasks*), alongside with other hyperparmeter tuning results.

# 6 Conclusion and discussions

In this paper, we proposed the first general framework to extend multi-objective optimization to the federated learning paradigm, which considers both objective and data heterogeneity. We showed that, even under objective and data heterogeneity, both of our proposed algorithms enjoy the same Pareto stationary convergence rate as their centralized counterparts. In our future work, we will go beyond the limitation in the analysis of MOO that an extra assumption on the stochastic gradients (and $\boldsymbol{\lambda}$). In this paper, we have proposed a weaker assumption (Assumption 4). We conjecture that using acceleration techniques, e.g., momentum, variance reduction, and regularization, could relax such assumption and achieve better convergence rate, which is a promising direction for future works. In addition, MOO in distributed learning gives rise to substantially expensive communication costs, which scales linearly with the number of clients and the number of objectives in each client. Developing communication-efficient MOO beyond typical gradient compression methods for distributed learning is also a promising direction for future works.

## Acknowledgments and Disclosure of Funding

This work has been supported in part by NSF grants CAREER CNS-2110259 and CNS-2112471.

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

# A   Gradient-based methods in MOO

(Stochastic) Gradient-based methods in MOO have attracted much attention owing to simple update rules and less intensive computation recently, thus rendering them perfect candidates to underpin MOO applications in deep learning under first-oracle. However, their theoretical understandings remain less explored relative to their counterparts of single objective optimization. Hence, we highlight the existing works and corresponding assumptions alongside with convergence metrics.

**Existing Works.** Various works managed to explore the convergence rates under different assumptions in strongly-convex, convex, and non-convex functions as listed in Table 4. Using full gradient, MGD [8] could achieve tight convergence rates in strongly-convex and non-convex cases, i.e., linear rate $\mathcal{O}(r^T), r \in (0,1)$ and sub-linear rate $\mathcal{O}(\frac{1}{T})$. However, it requires linear search of learning rate in the algorithm and sequence convergence ($\{\mathbf{x}_t\}$ converges to $\mathbf{x}_*$). The linear search of learning rate is a classic technique, but does not fits in gradient-based algorithms in deep learning. Moreover, sequence convergence assumption is a too strong assumption. With no local step, our FMGDA degenerates to MGD. As a result, our analysis also provide the same order convergence rates in both strongly-convex and non-convex functions while avoiding strong and unpractical assumptions. If using stochastic gradient, SMGD methods makes a further complicated case. The stochastic gradient noise would complicate the analysis and thus it is still unclear whether SMGD is guaranteed to converge. [9] provided convergence rate for SMGD but extra assumptions and/or unreasonably large batch requirements were needed. On the other hand, [9] and [18] showed that the common descent direction provided by SMGD method is likely to be a biased estimation, rendering non-convergence issues. Recently, by utilizing momentum, MoCo [19] and CR-MOGM [18] were proposed with corresponding convergence guarantees. [20] utilized direction-oriented approach by a preference direction. However, these analyses do not shed light on pure SMGD despite its widespread application.

**Assumptions.** When applying stochastic gradient to MOO, common descent direction estimation $\boldsymbol{\lambda}^T \nabla \mathbf{F}(\mathbf{x}, \xi)$ ($\mathbf{F}(\mathbf{x}) = [f_1(\mathbf{x}), \cdots, f_S(\mathbf{x})]$) is a biased estimation and thus rendering potential non-convergence issues [9, 18]. This is a limitation for SMGD. However, SMGD does work well with a wide range of applications in practice. Understanding under what conditions can SMGD have convergence guarantee is thus an important problem. [56] assumes convexity property(H5): $f(\mathbf{x}, \xi) - f(\mathbf{x}^*, \xi) \geq \frac{c}{2}\|\mathbf{x} - \mathbf{x}^*\|^2$ almost sure. [9] utilizes weaker assumptions but still needs first moment bound (Assumption 5.2(b)): $\mathbb{E}[\|\nabla f(\mathbf{x}, \xi) - \nabla f(\mathbf{x})\|] \leq \eta(a + b\|\nabla f(\mathbf{x})\|)$ and Lipschitz continuity of $\lambda$ (Assumption 5.4): $\|\boldsymbol{\lambda}_k - \boldsymbol{\lambda}_t\| \leq \beta \left\| \left[ (\nabla f_1(\mathbf{x}_k) - \nabla f_1(\mathbf{x}_t))^T, \ldots, (\nabla f_S(\mathbf{x}_k) - \nabla f_S(\mathbf{x}_t))^T \right] \right\|$.

In this paper, we use $(\alpha, \beta)$-Lipschitz continuous stochastic gradient (Assumption 4). In essence, we need the stochastic gradient estimation satisfying $\mathbb{E}[\|\nabla f(\mathbf{x}, \xi) - \nabla f(\mathbf{y}, \xi')\|^2] \leq \alpha\|\mathbf{x} - \mathbf{y}\|^2 + \beta\sigma^2$ for any two independent samples $\xi$ and $\xi'$. For the inequality $\mathbb{E}[\|\nabla f(\mathbf{x}, \xi) - \nabla f(\mathbf{y}, \xi')\|^2] \leq \alpha\|\mathbf{x} - \mathbf{y}\|^2 + \beta\sigma^2$ in Assumption 4, the notation $\sigma^2$ just represents a general positive constant. This $\sigma^2$ does not denote the variance of the stochastic gradient variance. Thus, this inequality does not

Table 3: List of key notation.

| Notation | Definition |
|---|---|
| $i$ | Client index |
| $M$ | Total number of clients |
| $s$ | Objective/task index |
| $S$ | Total number of Objectives/tasks |
| $S_i$ | Number of objectives/tasks of client $i$'s interest |
| $k$ | Local step index |
| $K$ | Total number of local steps |
| $t$ | Communication round index |
| $T$ | Total number of communication rounds |
| $\mathbf{x} \in \mathbb{R}^d$ | Global model parameters of FMOL in Problem (2) |
| $\mathbf{x}_0 \in \mathbb{R}^d$ | Initial solution of FMOL in Problem (2) |
| $\mathbf{x}_* \in \mathbb{R}^d$ | A Pareto optimal solution of FMOL in Problem (2) |
| $\eta_L$ | The learning rate on the client side |
| $\eta_t$ | The learning rate on the server side in round $t$ |

depend on the batch size of the stochastic gradient. More specifically, unlike the assumption in [9] that characterizes the difference between a stochastic gradient and its full gradient (hence depending on the batch size), our Assumption 4 only measures the average norm square of two stochastic gradient difference $\nabla f(\mathbf{x}, \xi) - \nabla f(\mathbf{y}, \xi^{'})$ given any two points $\mathbf{x}$ and $\mathbf{y}$ and any two samples $\xi$ and $\xi^{'}$. In other words, Assumption 4 does not involve any full gradient, and hence no dependence on batch size.

It is a natural extension of the classic Lipschitz continuous gradient assumption and could generalize existing assumptions.

1. If $\xi$ and $\xi^{'}$ are the whole dataset, by setting $\alpha = L^2$ and $\beta = 0$, $(\alpha, \beta)$-Lipschitz continuous stochastic gradient condition generalizes the traditional Lipschitz continuous gradient assumption $\|\nabla f(\mathbf{x}) - \nabla f(\mathbf{y})\| \le L\|\mathbf{x} - \mathbf{y}\|$.

2. If $\xi$ is one data sample, $\xi^{'}$ are the whole dataset and $\mathbf{x} = \mathbf{y}$, by setting $\alpha = 0$ and $\beta = 1$, $(\alpha, \beta)$-Lipschitz continuous stochastic gradient condition generalizes the traditional bounded variance assumption $\|\nabla f(\mathbf{x}, \xi) - \nabla f(\mathbf{x})\|^2 \le \sigma^2$.

3. If $\xi$ is one data sample, $\xi^{'}$ are the whole dataset and $\mathbf{x} = \mathbf{y}$, by setting $\beta = \alpha_k$, $(\alpha, \beta)$-Lipschitz continuous stochastic gradient condition generalizes the bound on the first moment assumption (assumption 5.2(b)) and bounded sets assumption (assumption 5.3) [9] ($\mathbb{E}[\|\nabla f(\mathbf{x}, \xi) - \nabla f(\mathbf{x})\|] \le \alpha_k(C_i + \hat{C}_i\|\nabla f_i(\mathbf{x}_k)\|)$ and $\|\nabla f_i(\mathbf{x})\| \le M_\nabla + L\Theta$).

**Metrics.** For strongly-convex functions, we use $\Delta_Q^t = \sum_{s \in [S]} \lambda_s^{t,*}[f_s(\mathbf{x}_t) - f_s(\mathbf{x}_*)]$ as the metrics. We note similar metrics are used in other works. For example, [9] uses $\min_{t=1,\dots,T} \sum_{s \in [S]} [\lambda_s^{t,*} f_s(\mathbf{x}_t) - \bar{\lambda}_T f_s(\mathbf{x}_*)]$ where $\bar{\lambda}_T = \sum_{t=1}^{T} \frac{t}{\sum_{t=1}^{T} t} \lambda_t$. Here $\lambda_s^{t,*}$ is calculated by the quadratic programming problem 3 with stochastic gradients. Rigorously speaking, the left-hand side is not guaranteed to be positive. But if we impose stronger assumptions as shown in [9, 56], we can have the same convergence metric as that in single objective optimization as an direct extension. In non-convex functions, $\left\|\bar{\mathbf{d}}_t\right\|^2$ are used as the metrics for FMOO, where $\bar{\mathbf{d}}_t = \boldsymbol{\lambda}_t^T \nabla \mathbf{F}(\mathbf{x}_t)$ and $\boldsymbol{\lambda}_t$ is calculated based on accumulated (stochastic) gradients $\Delta_t$. We note, directly extended from MOO [18, 19], $\mathbf{d}_t^* = \hat{\boldsymbol{\lambda}}_t^{*T} \nabla \mathbf{F}(\mathbf{x}_t)$ could also be used as the metrics in FMOO, where $\hat{\boldsymbol{\lambda}}_t^*$ is calculated based on full gradients $\nabla \mathbf{F}(\mathbf{x}_t)$. However, we prefer $\bar{\mathbf{d}}_t$ for the following reasons: i). For applying gradient descent with no local steps, $\bar{\mathbf{d}}_t$ degenerates to $\mathbf{d}_t^*$. ii). Clearly, $\|\bar{\mathbf{d}}_t\|^2 \ge \|\mathbf{d}_t^*\|^2$ as $\hat{\boldsymbol{\lambda}}_t^*$ is calculated based on gradients $\nabla \mathbf{F}(\mathbf{x}_t)$. Hence, $\|\bar{\mathbf{d}}_t\|^2$ is stronger convergence measure for FMOO. iii). $\boldsymbol{\lambda}_t$ is calculated in the algorithm and thus being more practical to use in practice, while $\hat{\boldsymbol{\lambda}}_t^*$ is unknown. Also, the convergence of $\bar{\mathbf{d}}_t$ implicitly indicates $\boldsymbol{\lambda}_t$ converges to $\hat{\boldsymbol{\lambda}}_t^*$.

## B  Proof of gradient descent type methods

For gradient descent type methods, each step utilizes a full gradient to update and the corresponding parameter $\lambda$ is deterministic. For clarity of notation, we drop $*$ for $\lambda$, that is, we use $\lambda_t^s$ to represent the solution of quadratic problem (Step 6 in the algorithm) for task $s$ in the $t$-th round.

**Lemma 1.** *Under bounded gradient assumption, the local model updates for any client $s$ could be bounded*

$$G_{s,i}^{t,k} = \|\mathbf{x}_{s,i}^{t,k} - \mathbf{x}_t\|^2 \le 4\eta_L^2 K^2 G^2, \tag{4}$$

$$H_{t,s} = \|\nabla f_s(\mathbf{x}_t) - \Delta_s^t\|^2 \le 4\eta_L^2 K^2 L^2 G^2. \tag{5}$$

*Proof.* For one task $s \in [S]$ and one client $i \in R_s$, the local update $\left\|\mathbf{x}_t - \mathbf{x}_{s,i}^{t,k}\right\|^2$ could be further bounded.

$$\left\|\mathbf{x}_t - \mathbf{x}_{s,i}^{t,k}\right\|^2 = \left\|\mathbf{x}_t - \mathbf{x}_{s,i}^{t,k-1} + \eta_L \nabla f_{s,i}(\mathbf{x}_{s,i}^{t,k-1})\right\|^2 \tag{6}$$

$$\le (1 + \frac{1}{K-1}) \left\|\mathbf{x}_t - \mathbf{x}_{s,i}^{t,k-1}\right\|^2 + \eta_L^2 K \left\|\nabla f_{s,i}(\mathbf{x}_{s,i}^{t,k-1})\right\|^2 \tag{7}$$

Table 4: Convergence rate (shaded parts are our results) for strongly-convex and non-convex functions, respectively:

| Methods | | Rate | | Assumption |
| Setting | Algorithm | SC | NC | |
| --- | --- | --- | --- | --- |
| Vanilla Gradient | MGD [8] | $\mathcal{O}(r^T), r \in (0,1)$ | $\mathcal{O}(\frac{1}{T})$ | Sequence convergence |
| | MGD | $\mathcal{O}(exp(-\mu T))$ | $\mathcal{O}(\frac{1}{T})$ | - |
| | SMGD [9] | $\mathcal{O}(\frac{1}{T})$ | - | Lipschitz continuity of $\lambda$ |
| | SMGD [56] | $\mathcal{O}(\frac{1}{T})$ | - | Convexity property |
| | SMGD [39] | - | $\mathcal{O}(\frac{1}{\sqrt{T}})$ | Given exact solution $\lambda^*$ |
| | SMGD | $\tilde{\mathcal{O}}(\frac{1}{T})$ | $\mathcal{O}(\frac{1}{\sqrt{T}})$ | Asm. 4 |
| Momentum | MoCo [19] | - | $\mathcal{O}(\frac{1}{\sqrt{T}})$ | - |
| | CR-MOGM [18] | - | $\mathcal{O}(\frac{1}{\sqrt{T}})$ | - |
| Federated Settings | FMGDA | $\mathcal{O}(exp(-\mu T))$ | $\mathcal{O}(\frac{1}{T})$ | - |
| | FSMGDA | $\tilde{\mathcal{O}}(\frac{1}{T})$ | $\mathcal{O}(\frac{1}{\sqrt{T}})$ | Asm. 4 |

**Assumptions.** Linear search [8]: stepsize linear search; sequence convergence [8]: $\{\mathbf{x}_t\}$ converges to $\mathbf{x}_*$; first moment bound (Asm. 5.2(b) [9]): $\mathbb{E}[\|\nabla f(\mathbf{x}, \xi) - \nabla f(\mathbf{x})\|] \leq \eta(a + b\|\nabla f(\mathbf{x})\|)$; Lipschitz continuity of $\lambda$ (Asm. 5.4 [9]): $\|\lambda_k - \lambda_s\| \leq \beta \|[(\nabla f_1(\mathbf{x}_k) - \nabla f_1(\mathbf{x}_t))^T, \ldots, (\nabla f_m(\mathbf{x}_k) - \nabla f_m(\mathbf{x}_t))^T]\|$; convexity property(H5) [56]: $f(\mathbf{x}, \xi) - f(\mathbf{x}^*, \xi) \geq \frac{c}{2}\|\mathbf{x} - \mathbf{x}^*\|^2$ almost sure; $(\alpha, \beta)$-Lipschitz continuous stochastic gradient (Asm. 4).

$$\leq (1 + \frac{1}{K-1}) \left\| \mathbf{x}_t - \mathbf{x}_{s,i}^{t,k-1} \right\|^2 + \eta_L^2 K G^2 \tag{8}$$

$$\leq \sum_{\tau \in [k-1]} \left(2\eta_L^2 K G^2\right) \left(1 + \frac{1}{K-1}\right)^{\tau} \tag{9}$$

$$\leq (K-1) \left[ \left(1 + \frac{1}{K-1}\right)^{K} - 1 \right] (\eta_L^2 K G^2) \tag{10}$$

$$\leq 4\eta_L^2 K^2 G^2, \tag{11}$$

where the first inequality comes from Young's inequality, the second inequality follows from bounded gradient assumption, and the last inequality follows if $\left(1 + \frac{1}{K-1}\right)^{K} - 1 \leq 4$ for $K > 1$.

We have the bound for local update for each task $s$, $H_{t,s}$, as follows:

$$H_{t,s} = \|\nabla f_s(\mathbf{x}_t) - \Delta_s^t\|^2 \tag{12}$$

$$= \left\| \frac{1}{K} \sum_{k \in [K]} \frac{1}{|R_s|} \sum_{i \in R_s} \left[ \nabla f_{s,i}(\mathbf{x}_t) - \nabla f_{s,i}(\mathbf{x}_{s,i}^{t,k}) \right] \right\|^2 \tag{13}$$

$$\leq \frac{1}{K} \sum_{k \in [K]} \frac{1}{|R_s|} \sum_{i \in R_s} \left\| \nabla f_{s,i}(\mathbf{x}_t) - \nabla f_{s,i}(\mathbf{x}_{s,i}^{t,k}) \right\|^2 \tag{14}$$

$$\leq \frac{1}{K} L^2 \sum_{k \in [K]} \frac{1}{|R_s|} \sum_{i \in R_s} \left\| \mathbf{x}_t - \mathbf{x}_{s,i}^{t,k} \right\|^2 \tag{15}$$

$$\leq 4\eta_L^2 K^2 L^2 G^2. \tag{16}$$

$\square$

**Lemma 2.** *For general L-smooth functions $\{f_s, s \in [S]\}$, choose the learning rate $\eta_t$ s.t. $\eta_t \leq \frac{3}{2(1+L)}$, the update $d_t$ of the algorithm satisfies:*

$$\frac{\eta_t}{4} \|\mathbf{d}_t\|^2 \leq -f_s(\mathbf{x}_{t+1}) + f_s(\mathbf{x}_t) + 6\eta_L^2 K^2 L^2 G^2 \tag{17}$$

*Proof.*

$$f_s(\mathbf{x}_{t+1}) \leq f_s(\mathbf{x}_t) + \langle \nabla f_s(\mathbf{x}_t), -\eta_t \mathbf{d}_t \rangle + \frac{1}{2} L \|\eta_t \mathbf{d}_t\|^2 \tag{18}$$

$$= f_s(\mathbf{x}_t) + \langle \nabla f_s(\mathbf{x}_t) - \Delta_s^t, -\eta_t \mathbf{d}_t \rangle - \eta_t \langle \Delta_s^t, \mathbf{d}_t \rangle + \frac{1}{2} L \|\eta_t \mathbf{d}_t\|^2 \tag{19}$$

$$\leq f_s(\mathbf{x}_t) + \langle \nabla f_s(\mathbf{x}_t) - \Delta_s^t, -\eta_t \mathbf{d}_t \rangle - \eta_t \|\mathbf{d}_t\|^2 + \frac{1}{2} L \|\eta_t \mathbf{d}_t\|^2 \tag{20}$$

$$\leq f_s(\mathbf{x}_t) + \frac{1}{2} \|\nabla f_s(\mathbf{x}_t) - \Delta_s^t\|^2 + \frac{1}{2} \eta_t^2 \|\mathbf{d}_t\|^2 - \eta_t \|\mathbf{d}_t\|^2 + \frac{1}{2} L \eta_t^2 \|\mathbf{d}_t\|^2 \tag{21}$$

$$= f_s(\mathbf{x}_t) + \frac{1}{2} \|\nabla f_s(\mathbf{x}_t) - \Delta_s^t\|^2 - \eta_t \left(1 - \frac{1}{2}\eta_t - \frac{1}{2}L\eta_t\right) \|\mathbf{d}_t\|^2 \tag{22}$$

$$\leq f_s(\mathbf{x}_t) + 2\eta_L^2 K^2 L^2 G^2 - \eta_t \left(1 - \frac{1}{2}\eta_t - \frac{1}{2}L\eta_t\right) \|\mathbf{d}_t\|^2. \tag{23}$$

The third inequality follows from $\langle \Delta_s^t, \mathbf{d}_t \rangle \geq \|\mathbf{d}_t\|^2$ since $\mathbf{d}_t$ is a general solution in the convex hull of the family of vectors $\{\Delta_s^t, s \in [S]\}$ (see Lemma 2.1 [15]). Here $\mathbf{d}_t = \sum_{s \in [S]} \lambda_s^{t,*} \Delta_s^t$ and $\lambda_s^{t,*}$ is calculated by $\Delta_s^t$, but we drop the $*$ of $\lambda$ for simplicity.

By setting $\left(1 - \frac{1}{2}\eta_t - \frac{1}{2}L\eta_t\right) \geq \frac{1}{4}$, that is, $\eta_t \leq \frac{3}{2(1+L)}$, we have

$$\frac{\eta_t}{4} \|\mathbf{d}_t\|^2 \leq -f_s(\mathbf{x}_{t+1}) + f_s(\mathbf{x}_t) + 2\eta_L^2 K^2 L^2 G^2. \tag{24}$$

$\square$

## B.1 Strongly Convex Functions

**Theorem 3** (FMGDA for $\mu$-Strongly Convex FMOL). *Let $\eta_t = \eta$ such that $\eta \leq \frac{3}{2(1+L)}$, $\eta \leq \frac{1}{2L+\mu}$ and $\eta \geq \frac{1}{\mu T}$. Under Assumptions 1- 3, pick $\mathbf{x}_t$ as the final output of the FMGDA algorithm with weights $w_t = (1 - \frac{\mu\eta}{2})^{1-t}$. Then, it holds that $\mathbb{E}[\Delta_Q^t] \leq \|\mathbf{x}_0 - \mathbf{x}_*\|^2 \mu \exp(-\frac{\eta\mu T}{2}) + \delta$, where $\Delta_Q^t \triangleq \sum_{s \in [S]} \lambda_s^{t,*} [f_s(\mathbf{x}_t) - f_s(\mathbf{x}_*)]$ and $\delta = \frac{8\eta_L^2 K^2 L^2 G^2 S^2}{\mu} + 2\eta_L^2 K^2 L^2 G^2$.*

*Proof.*

$$f_s(\mathbf{x}_{t+1}) \leq f_s(\mathbf{x}_t) + \langle \nabla f_s(\mathbf{x}_t), -\eta_t \mathbf{d}_t \rangle + \frac{1}{2} L \|\eta_t \mathbf{d}_t\|^2 \tag{25}$$

$$\leq f_s(\mathbf{x}_*) + \langle \nabla f_s(\mathbf{x}_t), \mathbf{x}_t - \mathbf{x}_* \rangle - \frac{\mu}{2} \|\mathbf{x}_t - \mathbf{x}_*\|^2 \tag{26}$$

$$+ \langle \nabla f_s(\mathbf{x}_t), -\eta_t \mathbf{d}_t \rangle + \frac{1}{2} L \|\eta_t \mathbf{d}_t\|^2, \tag{27}$$

where the first inequality is due to $L$-smoothness, the second inequality follows from $\mu$-strongly convex.

$$\sum_{s \in [S]} \lambda_t^s [f_s(\mathbf{x}_{t+1}) - f_s(\mathbf{x}_*)] \tag{28}$$

$$\leq \left\langle \sum_{s \in [S]} \lambda_t^s \nabla f_s(\mathbf{x}_t), \mathbf{x}_t - \mathbf{x}_* \right\rangle - \frac{\mu}{2} \|\mathbf{x}_t - \mathbf{x}_*\|^2 + \left\langle \sum_{s \in [S]} \lambda_t^s \nabla f_s(\mathbf{x}_t), -\eta_t \mathbf{d}_t \right\rangle + \frac{1}{2} L \|\eta_t \mathbf{d}_t\|^2 \tag{29}$$

$$= \left\langle \sum_{s \in [S]} \lambda_t^s \nabla f_s(\mathbf{x}_t), \mathbf{x}_t - \mathbf{x}_* - \eta_t \mathbf{d}_t \right\rangle - \frac{\mu}{2} \|\mathbf{x}_t - \mathbf{x}_*\|^2 + \frac{1}{2} L \|\eta_t \mathbf{d}_t\|^2 \tag{30}$$

$$= \langle \mathbf{d}_t, \mathbf{x}_t - \mathbf{x}_* - \eta_t \mathbf{d}_t \rangle - \frac{\mu}{2} \|\mathbf{x}_t - \mathbf{x}_*\|^2 + \frac{1}{2} L \|\eta_t \mathbf{d}_t\|^2 + \left\langle \sum_{s \in [S]} \lambda_t^s \nabla f_s(\mathbf{x}_t) - \mathbf{d}_t, \mathbf{x}_t - \mathbf{x}_* - \eta_t \mathbf{d}_t \right\rangle \tag{31}$$

$$= \langle \mathbf{d}_t, \mathbf{x}_t - \mathbf{x}_* \rangle - \eta_t \|\mathbf{d}_t\|^2 - \frac{\mu}{2}\|\mathbf{x}_t - \mathbf{x}_*\|^2 + \frac{1}{2}L\eta_t^2\|\mathbf{d}_t\|^2 + \left\langle \sum_{s\in[S]} \lambda_t^s \nabla f_s(\mathbf{x}_t) - \mathbf{d}_t, \mathbf{x}_{t+1} - \mathbf{x}_* \right\rangle \tag{32}$$

$$\leq \frac{1}{2\eta_t}\left(\|\mathbf{x}_t - \mathbf{x}_*\|^2 - \|\mathbf{x}_{t+1} - \mathbf{x}_*\|^2\right) - \frac{1}{2}\eta_t\|\mathbf{d}_t\|^2 - \frac{\mu}{2}\|\mathbf{x}_t - \mathbf{x}_*\|^2 + \frac{1}{2}L\eta_t^2\|\mathbf{d}_t\|^2 \tag{33}$$

$$+ \frac{1}{4\epsilon} \underbrace{\left\| \sum_{s\in[S]} \lambda_t^s \nabla f_s(\mathbf{x}_t) - \mathbf{d}_t \right\|^2}_{H_t} + \epsilon \|\mathbf{x}_{t+1} - \mathbf{x}_*\|^2 \tag{34}$$

$$\leq \frac{1}{2\eta_t}\left(\|\mathbf{x}_t - \mathbf{x}_*\|^2 - \|\mathbf{x}_{t+1} - \mathbf{x}_*\|^2\right) - \frac{1}{2}\eta_t\|\mathbf{d}_t\|^2 - \frac{\mu}{2}\|\mathbf{x}_t - \mathbf{x}_*\|^2 + \frac{1}{2}L\eta_t^2\|d_t\|^2 \tag{35}$$

$$+ \frac{1}{4\epsilon}H_t + \epsilon\left(2\|\mathbf{x}_t - \mathbf{x}_*\|^2 + 2\eta_t^2\|\mathbf{d}_t\|^2\right) \tag{36}$$

$$\leq \frac{1}{2\eta_t}\left((1 - \frac{\mu}{2}\eta_t)\|\mathbf{x}_t - \mathbf{x}_*\|^2 - \|\mathbf{x}_{t+1} - \mathbf{x}_*\|^2\right) - \left(\frac{1}{2}\eta_t - \frac{1}{2}L\eta_t^2 - \frac{\mu}{4}\eta_t^2\right)\|\mathbf{d}_t\|^2 + \frac{2}{\mu}H_t, \tag{37}$$

where $\|\mathbf{x}_t - \mathbf{x}_*\|^2 - \|\mathbf{x}_{t+1} - \mathbf{x}_*\|^2 = -\eta_t^2\|\mathbf{d}_t\|^2 + 2\langle \eta_t \mathbf{d}_t, \mathbf{x}_t - \mathbf{x}_* \rangle$, and we choose $\epsilon = \frac{\mu}{8}$ in the last inequality.

From Lemma 2, it is clear that

$$|f_s(\mathbf{x}_{t+1}) - f_s(\mathbf{x}_t)| \leq |2\eta_L^2 K^2 L^2 G^2 - \frac{\eta_t}{4}\|\mathbf{d}_t\|^2| \tag{38}$$

$$\leq 2\eta_L^2 K^2 L^2 G^2 + \frac{\eta_t}{4}\|\mathbf{d}_t\|^2. \tag{39}$$

$$\Delta_Q^t = \sum_{s\in[S]} \lambda_t^s \left[f_s(\mathbf{x}_t) - f_s(\mathbf{x}_*)\right] \leq \sum_{s\in[S]} \lambda_t^s \left[f_s(\mathbf{x}_{t+1}) - f_s(\mathbf{x}_*)\right] + |f_s(\mathbf{x}_{t+1}) - f_s(\mathbf{x}_t)| \tag{40}$$

$$\leq \frac{1}{2\eta_t}\left((1 - \frac{\mu}{2}\eta_t)\|\mathbf{x}_t - \mathbf{x}_*\|^2 - \|\mathbf{x}_{t+1} - \mathbf{x}_*\|^2\right) - \left(\frac{1}{4}\eta_t - \frac{1}{2}L\eta_t^2 - \frac{\mu}{4}\eta_t^2\right)\|\mathbf{d}_t\|^2 + \frac{2}{\mu}H_t + 2\eta_L^2 K^2 L^2 G^2. \tag{41}$$

$$H_t = \left\| \sum_{s\in[S]} \lambda_t^s \nabla f_s(\mathbf{x}_t) - \mathbf{d}_t \right\|^2 \tag{42}$$

$$\leq S \sum_{s\in[S]} (\lambda_t^s)^2 H_{t,s} \tag{43}$$

$$\leq 4\eta_L^2 K^2 L^2 G^2 S^2. \tag{44}$$

By setting $\eta_t \leq \frac{1}{2L+\mu}$, we have

$$\Delta_Q^t = \sum_{s\in[S]} \lambda_t^s \left[f_s(\mathbf{x}_{t+1}) - f_s(\mathbf{x}_*)\right] \tag{45}$$

$$\leq \frac{1}{2\eta_t}\left((1 - \frac{\mu}{2}\eta_t)\|\mathbf{x}_t - \mathbf{x}_*\|^2 - \|\mathbf{x}_{t+1} - \mathbf{x}_*\|^2\right) + \underbrace{\frac{8\eta_L^2 K^2 L^2 G^2 S^2}{\mu} + 2\eta_L^2 K^2 L^2 G^2}_{\delta}. \tag{46}$$

Averaging using weight $w_t = (1 - \frac{\mu\eta}{2})^{1-t}$ and using such weight to pick output $\mathbf{x}$. By using Lemma 1 in [27] with $\eta \geq \frac{1}{uR}$, we ahve

$$\mathbb{E}[\Delta_Q] \leq \|\mathbf{x}_0 - \mathbf{x}_*\|^2 \mu \exp(-\frac{\eta\mu T}{2}) + \delta \tag{47}$$

$$= \mathcal{O}(\mu \exp(-\mu T)) + \mathcal{O}(\delta). \tag{48}$$

If we set $\eta_L$ sufficiently small such that $\delta = \mathcal{O}(\mu \exp(-\mu T))$, then we have the convergence rate $\mathbb{E}[\Delta_Q] = \mathcal{O}(\mu \exp(-\mu T))$. $\qquad \square$

## B.2   Non-Convex Functions

**Theorem 1** (FMGDA for Non-convex FMOL). *Let $\eta_t = \eta \leq \frac{3}{2(1+L)}$. Under Assumptions 1 and 2, if at least one function $f_s, s \in [S]$ is bounded from below by $f_s^{\min}$, then the sequence $\{\mathbf{x}_t\}$ output by FMGDA satisfies:* $\min_{t \in [T]} \|\bar{\mathbf{d}}_t\|^2 \leq \frac{16(f_s^0 - f_s^{\min})}{T\eta} + \delta$, where $\delta \triangleq \frac{16\eta_L^2 K^2 L^2 G^2 (1+S^2)}{\eta}$.

*Proof.* From Lemma 2, we have

$$\frac{\eta_t}{4}\|\mathbf{d}_t\|^2 \leq -f_s(\mathbf{x}_{t+1}) + f_s(\mathbf{x}_t) + 2\eta_L^2 K^2 L^2 G^2. \tag{49}$$

With constant learning rate $\eta_t = \eta$,

$$\frac{1}{T}\sum_{t \in [T]} \|\mathbf{d}_t\|^2 \leq \frac{4(f_s^0 - f_s^{min})}{T\eta} + \frac{8\eta_L^2 K^2 L^2 G^2}{\eta}. \tag{50}$$

Note that $\left\|\bar{\mathbf{d}}_t\right\|^2$ are used as the metrics for FMOO, where $\bar{\mathbf{d}}_t = \boldsymbol{\lambda}_t^T \nabla(\text{Diag}(\mathbf{F}\mathbf{A}^\top))$ and $\boldsymbol{\lambda}_t$ is calculated based on accumulated (stochastic) gradients $\Delta_t$. Then we have

$$\left\|\bar{\mathbf{d}}_t\right\|^2 \leq 2 \left\|\sum_{s \in [S]} \lambda_t^s \nabla f_s(\mathbf{x}_t) - \mathbf{d}_t\right\|^2 + 2\|\mathbf{d}_t\|^2. \tag{51}$$

Thus,

$$\frac{1}{T}\sum_{t \in [T]} \|\bar{\mathbf{d}}_t\|^2 \leq \frac{16(f_s^0 - f_s^{min})}{T\eta} + \frac{16\eta_L^2 K^2 L^2 G^2 (1+S^2)}{\eta}. \tag{52}$$

With constant learning rate $\eta$ and local learning rate $\eta_L = \mathcal{O}(\frac{1}{\sqrt{T}KLGS})$, we have

$$\frac{1}{T}\sum_{t \in [T]} \|\bar{\mathbf{d}}_t\|^2 \leq \mathcal{O}(\frac{1}{T}) \tag{53}$$

$\qquad \square$

## C   Proof of stochastic gradient descent type methods

For stochastic gradient descent type methods, each step utilizes a stochastic gradient to update and the corresponding parameter $\lambda$ is stochastic, depending on the random samples in each client. For clarity of notation, we drop $*$ for $\lambda$, that is, we use $\lambda_t^s$ to represent the solution of quadratic problem (Step 6 in the algorithm) for task $s$ in the $t$-th round.

**Lemma 3.** *Under bounded stochastic gradient assumption, the local model updates could be bounded*

$$G_{s,i}^{t,k} = \mathbb{E}\|\mathbf{x}_{s,i}^{t,k} - \mathbf{x}_t\|^2 \leq 6\eta_L^2 k^2 \|\nabla f_{s,i}(\mathbf{x}_t)\|^2, \tag{54}$$

$$\mathbb{E}\left\|\sum_{s \in [S]} \lambda_s^t \Delta_s^t\right\|^2 \leq S^2 D^2. \tag{55}$$

*Further with assumption 4, we have*

$$H_{t,s} = \mathbb{E}\left\|\nabla f_s(\mathbf{x}_t, \xi_t) - \Delta_s^t\right\|^2 \leq \alpha \eta_L^2 K^2 D^2 + \beta \sigma^2. \tag{56}$$

*Proof.* For one task $s \in [S]$ and one client $i \in R_s$, the local update $\left\| \mathbf{x}_t - \mathbf{x}_{s,i}^{t,k} \right\|^2$ could be further bounded.

$$G_{s,i}^{t,k} = \mathbb{E} \left\| \mathbf{x}_t - \mathbf{x}_{s,i}^{t,k} \right\|^2 \tag{57}$$

$$= \mathbb{E} \left\| \sum_{\tau \in [k]} \eta_L \nabla f_{s,i}(\mathbf{x}_{s,i}^{t,\tau}, \xi_{s,i}^{t,\tau}) \right\|^2 \tag{58}$$

$$\leq \eta_L^2 k^2 D^2. \tag{59}$$

$$\mathbb{E} \left\| \sum_{s \in [S]} \lambda_s^t \Delta_s^t \right\|^2 \leq S \sum_{s \in [S]} \mathbb{E} \left[ (\lambda_s^t)^2 \left\| \Delta_s^t \right\|^2 \right] \tag{60}$$

$$\leq S \sum_{s \in [S]} \mathbb{E} \left[ \left\| \Delta_s^t \right\|^2 \right] \tag{61}$$

$$\leq S \sum_{s \in [S]} \mathbb{E} \left\| \frac{1}{R_s} \sum_{i \in R_s} \frac{1}{K} \sum_{\tau \in [K]} \nabla f_{s,i}(\mathbf{x}_{s,i}^{t,\tau}, \xi_{s,i}^{t,\tau}) \right\|^2 \tag{62}$$

$$\leq S \sum_{s \in [S]} \frac{1}{R_s} \sum_{i \in R_s} \frac{1}{K} \sum_{\tau \in [K]} \mathbb{E} \left\| \nabla f_{s,i}(\mathbf{x}_{s,i}^{t,\tau}, \xi_{s,i}^{t,\tau}) \right\|^2 \tag{63}$$

$$\leq S^2 D^2. \tag{64}$$

$$H_{t,s} = \mathbb{E} \left\| \nabla f_s(\mathbf{x}_t, \xi_t) - \Delta_s^t \right\|^2 \tag{65}$$

$$\leq \mathbb{E} \left\| \frac{1}{K} \sum_{k \in [K]} \frac{1}{|R_s|} \sum_{i \in R_s} \left( \nabla f_{s,i}(\mathbf{x}_t, \xi_t) - \nabla f_{s,i}(\mathbf{x}_{s,i}^{t,k}, \xi_{s,i}^{t,k}) \right) \right\|^2 \tag{66}$$

$$\leq \frac{1}{K} \sum_{k \in [K]} \frac{1}{|R_s|} \sum_{i \in R_s} \mathbb{E} \left\| \nabla f_{s,i}(\mathbf{x}_t, \xi_t) - \nabla f_{s,i}(\mathbf{x}_{s,i}^{t,k}, \xi_{s,i}^{t,k}) \right\|^2 \tag{67}$$

$$\leq \frac{1}{K} \sum_{k \in [K]} \frac{1}{|R_s|} \sum_{i \in R_s} \left( \alpha \mathbb{E} \| \mathbf{x}_t - \mathbf{x}_{s,i}^{t,k} \|^2 + \beta \sigma^2 \right) \tag{68}$$

$$\leq \alpha \eta_L^2 K^2 D^2 + \beta \sigma^2. \tag{69}$$

$\square$

## C.1 Strongly Convex Functions

**Theorem 7** (FSMGDA for $\mu$-Strongly Convex FMOL). *Let $\eta_t = \eta = \Omega(\frac{1}{\mu T})$. Under Assumptions 3, 5 and 6, pick $\mathbf{x}_t$ as the final output of the FSMGDA algorithm with weight $w_t = (1 - \frac{\mu \eta}{2})^{1-t}$. Then, it holds that: $\mathbb{E}[\Delta_Q^t] \leq \|\mathbf{x}_0 - \mathbf{x}_*\|^2 \mu \exp(-\frac{\eta}{2}\mu T) + \delta$, where $\Delta_Q^t = \sum_{s \in [S]} \lambda_s^{t,*} [f_s(\mathbf{x}_t) - f_s(\mathbf{x}_*)]$ and $\delta = \frac{1}{\mu} S^2 (\alpha \eta_L^2 K^2 D^2 + \beta \sigma^2) + \frac{\eta S^2 D^2}{2}.$*

*Proof.* Taking expectation over random samples conditioning on $\mathbf{x}_t$, we have

$$\mathbb{E} \| \mathbf{x}_{t+1} - \mathbf{x}_* \|^2 = \mathbb{E} \left\| \mathbf{x}_t - \eta_t \sum_{s \in [S]} \lambda_s^t \Delta_s^t - x_* \right\|^2 \tag{70}$$

$$= \|\mathbf{x}_t - \mathbf{x}_*\|^2 - \mathbb{E}\left\langle \mathbf{x}_t - \mathbf{x}_*, 2\eta_t \sum_{s \in [S]} \lambda_s^t \Delta_s^t \right\rangle + \mathbb{E}\left\| \eta_t \sum_{s \in [S]} \lambda_s^t \Delta_s^t \right\|^2 \tag{71}$$

$$= \|\mathbf{x}_t - \mathbf{x}_*\|^2 - \mathbb{E}\left\langle \mathbf{x}_t - \mathbf{x}_*, 2\eta_t \sum_{s \in [S]} \lambda_s^t \nabla f_s(\mathbf{x}_t, \xi_t) \right\rangle \tag{72}$$

$$+ \mathbb{E}\left\langle \mathbf{x}_t - \mathbf{x}_*, 2\eta_t \sum_{s \in [S]} \lambda_s^t (\nabla f_s(\mathbf{x}_t, \xi_t) - \Delta_s^t) \right\rangle + \mathbb{E}\left\| \eta_t \sum_{s \in [S]} \lambda_s^t \Delta_s^t \right\|^2 \tag{73}$$

$$= \|\mathbf{x}_t - \mathbf{x}_*\|^2 - \left\langle \mathbf{x}_t - \mathbf{x}_*, 2\eta_t \sum_{s \in [S]} \mathbb{E}[\lambda_s^t] \nabla f_s(\mathbf{x}_t) \right\rangle \tag{74}$$

$$+ \mathbb{E}\left\langle \mathbf{x}_t - \mathbf{x}_*, 2\eta_t \sum_{s \in [S]} \lambda_s^t (\nabla f_s(\mathbf{x}_t, \xi_t) - \Delta_s^t) \right\rangle + \mathbb{E}\left\| \eta_t \sum_{s \in [S]} \lambda_s^t \Delta_s^t \right\|^2 \tag{75}$$

$$\leq \|\mathbf{x}_t - \mathbf{x}_*\|^2 - 2\eta_t \left( \frac{\mu}{2} \|\mathbf{x}_t - \mathbf{x}_*\|^2 + \sum_{s \in [S]} \mathbb{E}[\lambda_s^t](f_s(\mathbf{x}_t) - f_s(\mathbf{x}_*)) \right) + \epsilon \|\mathbf{x}_t - \mathbf{x}_*\|^2 \tag{76}$$

$$+ \frac{1}{4\epsilon} 4\eta_t^2 \mathbb{E}\left\| \sum_{s \in [S]} \lambda_s^t (\nabla f_s(\mathbf{x}_t, \xi_t) - \Delta_s^t) \right\|^2 + \eta_t^2 \mathbb{E}\left\| \sum_{s \in [S]} \lambda_s^t \Delta_s^t \right\|^2 \tag{77}$$

$$\leq \|\mathbf{x}_t - \mathbf{x}_*\|^2 - 2\eta_t \left( \frac{\mu}{2} \|\mathbf{x}_t - \mathbf{x}_*\|^2 + \sum_{s \in [S]} \mathbb{E}[\lambda_s^t](f_s(\mathbf{x}_t) - f_s(\mathbf{x}_*)) \right) + \epsilon \|\mathbf{x}_t - \mathbf{x}_*\|^2 \tag{78}$$

$$+ \frac{1}{4\epsilon} 4\eta_t^2 S \sum_{s \in [S]} \mathbb{E}\left[ (\lambda_s^t)^2 \left\| (\nabla f_s(\mathbf{x}_t, \xi_t) - \Delta_s^t) \right\|^2 \right] + \eta_t^2 \mathbb{E}\left\| \sum_{s \in [S]} \lambda_s^t \Delta_s^t \right\|^2 \tag{79}$$

$$\leq \|\mathbf{x}_t - \mathbf{x}_*\|^2 - 2\eta_t \left( \frac{\mu}{2} \|\mathbf{x}_t - \mathbf{x}_*\|^2 + \sum_{s \in [S]} \mathbb{E}[\lambda_s^t](f_s(\mathbf{x}_t) - f_s(\mathbf{x}_*)) \right) + \epsilon \|\mathbf{x}_t - \mathbf{x}_*\|^2 \tag{80}$$

$$+ \frac{1}{4\epsilon} 4\eta_t^2 S \sum_{s \in [S]} \mathbb{E}\left\| \nabla f_s(\mathbf{x}_t, \xi_t) - \Delta_s^t \right\|^2 + \eta_t^2 \mathbb{E}\left\| \sum_{s \in [S]} \lambda_s^t \Delta_s^t \right\|^2 \tag{81}$$

$$\leq \|\mathbf{x}_t - \mathbf{x}_*\|^2 - 2\eta_t \left( \frac{\mu}{2} \|\mathbf{x}_t - \mathbf{x}_*\|^2 + \sum_{s \in [S]} \mathbb{E}[\lambda_s^t](f_s(\mathbf{x}_t) - f_s(\mathbf{x}_*)) \right) + \epsilon \|\mathbf{x}_t - \mathbf{x}_*\|^2 \tag{82}$$

$$+ \frac{1}{4\epsilon} 4\eta_t^2 S^2 (\alpha \eta_L^2 K^2 D^2 + \beta \sigma^2) + \eta_t^2 S^2 D^2 \tag{83}$$

$$\leq (1 - \frac{\eta_t \mu}{2}) \|\mathbf{x}_t - \mathbf{x}_*\|^2 - 2\eta_t \left( \sum_{s \in [S]} \mathbb{E}[\lambda_s^t](f_s(\mathbf{x}_t) - f_s(\mathbf{x}_*)) \right) \tag{84}$$

$$+ \frac{2}{\mu} \eta_t S^2 (\alpha \eta_L^2 K^2 D^2 + \beta \sigma^2) + \eta_t^2 S^2 D^2, \tag{85}$$

where the first equality is due to strongly-convex objective functions, and we set $\epsilon = \frac{\eta_t \mu}{2}$.

$$\sum_{s \in [S]} \mathbb{E}[\lambda_s^t](f_s(\mathbf{x}) - f_s(\mathbf{x}_*)) \leq \frac{1}{2\eta_t}(1 - \frac{\eta_t \mu}{2}) \|\mathbf{x}_t - \mathbf{x}_*\|^2 - \frac{1}{2\eta_t} \|\mathbf{x}_{t+1} - \mathbf{x}_*\|^2 \tag{86}$$

$$+ \frac{1}{\mu} S^2 (\alpha \eta_L^2 K^2 D^2 + \beta \sigma^2) + \underbrace{\frac{\eta_t S^2 D^2}{2}}_{\delta} \tag{87}$$

Averaging using weight $w_t = (1 - \frac{\mu \eta_t}{2})^{1-t}$ and using such weight to pick output $\mathbf{x}$. By using Lemma 1 in [27] with constant learning rate $\eta_t = \eta = \Omega(\frac{1}{\mu T})$, we have

$$\mathbb{E}[\Delta_Q] \le \|\mathbf{x}_0 - \mathbf{x}_*\|^2 \mu \exp(-\frac{\eta}{2} \mu T) + \mathcal{O}(\delta) \tag{88}$$

where $\delta = \frac{1}{\mu} S^2 (\alpha \eta_L^2 K^2 D^2 + \beta \sigma^2) + \frac{\eta S^2 D^2}{2}$.

By letting $\beta = \eta$, $\eta_L = \mathcal{O}(\frac{1}{\sqrt{T}})$ and $\eta = \Theta(\frac{\log(\max(1, \mu^2 T))}{\mu T})$,

$$\mathbb{E}[\Delta_Q] \le \tilde{\mathcal{O}}(\frac{1}{T}). \tag{89}$$

$\square$

## C.2 Non-convex Functions

**Theorem 5** (FSMGDA for Non-convex FMOL). *Let $\eta_t = \eta \le \frac{3}{2(1+L)}$. Under Assumptions 4–6, if an objective $f_s$ is bounded from below by $f_s^{\min}$, then the sequence $\{\mathbf{x}_t\}$ output by FSMGDA satisfies:* $\min_{t \in [T]} \mathbb{E} \left\| \bar{\mathbf{d}}_t \right\|^2 \le \frac{8(f_s^0 - f_s^{\min})}{\eta T} + \delta$, *where $\delta = (2S^2 + 4)(\alpha \eta_L^2 K^2 D^2 + \beta \sigma^2)$.*

*Proof.* Similar to Lemma 2 and taking expectation on the random data samples conditioning on $\mathbf{x}_t$, we have

$$\mathbb{E} f_s(\mathbf{x}_{t+1}) \le f_s(\mathbf{x}_t) + \mathbb{E} \langle \nabla f_s(\mathbf{x}_t), -\eta_t \mathbf{d}_t \rangle + \frac{1}{2} L \mathbb{E} \|\eta_t \mathbf{d}_t\|^2 \tag{90}$$

$$= f_s(\mathbf{x}_t) + \mathbb{E} \langle \nabla f_s(\mathbf{x}_t) - \Delta_s^t, -\eta_t \mathbf{d}_t \rangle - \eta_t \mathbb{E} \langle \Delta_s^t, \mathbf{d}_t \rangle + \frac{1}{2} L \mathbb{E} \|\eta_t \mathbf{d}_t\|^2 \tag{91}$$

$$\le f_s(\mathbf{x}_t) + \mathbb{E} \langle \nabla f_s(\mathbf{x}_t) - \Delta_s^t, -\eta_t \mathbf{d}_t \rangle - \eta_t \mathbb{E} \|\mathbf{d}_t\|^2 + \frac{1}{2} L \mathbb{E} \|\eta_t \mathbf{d}_t\|^2 \tag{92}$$

$$\le f_s(\mathbf{x}_t) + \frac{1}{2} \mathbb{E} \|\nabla f_s(\mathbf{x}_t) - \Delta_s^t\|^2 + \frac{1}{2} \eta_t^2 \mathbb{E} \|\mathbf{d}_t\|^2 - \eta_t \mathbb{E} \|\mathbf{d}_t\|^2 + \frac{1}{2} L \mathbb{E} \eta_t^2 \|\mathbf{d}_t\|^2 \tag{93}$$

$$= f_s(\mathbf{x}_t) + \frac{1}{2} \mathbb{E} \|\nabla f_s(\mathbf{x}_t) - \Delta_s^t\|^2 - \eta_t \left(1 - \frac{1}{2} \eta_t - \frac{1}{2} L \eta_t\right) \mathbb{E} \|\mathbf{d}_t\|^2, \tag{94}$$

where $\mathbf{d}_t = \sum_{s \in [S]} \lambda_s^{t,*} \Delta_s^t$ and $\lambda_s^{t,*}$ is calculated by the accumulated stochastic gradients $\Delta_s^t, s \in [S]$, but we drop the $*$ of $\lambda$ for simplicity.

With $\eta_t \le \frac{3}{2(1+L)}$, we have

$$\frac{\eta_t}{4} \mathbb{E} \|\mathbf{d}_t\|^2 \le -f_s(\mathbf{x}_{t+1}) + f_s(\mathbf{x}_t) + \frac{1}{2} \mathbb{E} \|\nabla f_s(\mathbf{x}_t) - \Delta_s^t\|^2 \tag{95}$$

$$\le -f_s(\mathbf{x}_{t+1}) + f_s(\mathbf{x}_t) + \frac{1}{2} (\alpha \eta_L^2 K^2 D^2 + \beta \sigma^2) \tag{96}$$

With constant learning rate $\eta_t = \eta$,

$$\frac{1}{T} \sum_{t \in [T]} \mathbb{E} \|\mathbf{d}_t\|^2 \le \frac{4(f_s(\mathbf{x}_1) - \mathbb{E} f_s(\mathbf{x}_{T+1}))}{\eta T} + 2(\alpha \eta_L^2 K^2 D^2 + \beta \sigma^2) \tag{97}$$

Note that we want to use $\left\| \bar{\mathbf{d}}_t \right\|^2$ are used as the metrics, where $\bar{\mathbf{d}}_t = \boldsymbol{\lambda}_t^T \nabla(\text{Diag}(\mathbf{FA}^\top))$ and $\boldsymbol{\lambda}_t$ is calculated based on accumulated (stochastic) gradients $\Delta_t$. Then we have

$$\left\| \bar{\mathbf{d}}_t \right\|^2 \le 2 \left\| \sum_{s \in [S]} \lambda_t^s \nabla f_s(\mathbf{x}_t) - \mathbf{d}_t \right\|^2 + 2 \left\| \mathbf{d}_t \right\|^2. \tag{98}$$

With constant learning rate $\eta_t = \eta$ and averaging from $T$ communication rounds, we have

$$\frac{1}{T}\sum_{t\in[T]}\mathbb{E}\left\|\bar{\mathbf{d}}_t\right\|^2 \le \frac{1}{T}\sum_{t\in[T]} 2\mathbb{E}\left\|\sum_{s\in[S]}\lambda_t^s\nabla f_s(\mathbf{x}_t) - \mathbf{d}_t\right\|^2 + \frac{1}{T}\sum_{t\in[T]} 2\mathbb{E}\left\|\mathbf{d}_t\right\|^2 \tag{99}$$

$$\le \frac{1}{T}\sum_{t\in[T]} 2S\mathbb{E}\sum_{s\in[S]}\left\|\lambda_t^s(\nabla f_s(\mathbf{x}_t) - \Delta_s^t)\right\|^2 + \frac{1}{T}\sum_{t\in[T]} 2\mathbb{E}\left\|\mathbf{d}_t\right\|^2 \tag{100}$$

$$\le \frac{8\left(f_s(\mathbf{x}_1) - \mathbb{E}f_s(\mathbf{x}_{T+1})\right)}{\eta T} + (2S^2 + 4)(\alpha\eta_L^2 K^2 D^2 + \beta\sigma^2) \tag{101}$$

With constant learning rate $\eta = \frac{1}{\sqrt{T}}$, local learning rate $\eta_L = \mathcal{O}(\frac{1}{T^{1/4}})$ and $\beta = \eta$,

$$\frac{1}{T}\sum_{t\in[T]}\mathbb{E}\left\|\bar{\mathbf{d}}_t\right\|^2 = \mathcal{O}(\frac{1}{\sqrt{T}}). \tag{102}$$

$\square$

## D    Further Experiments and Additional Results

In the following, we provide the detailed machine learning models for our experiments:

**1) MultiMNIST Datasets and Learning Tasks:** We test the convergence performance of our algorithms using the "MultiMNIST" dataset [53], which is a multi-task learning version of the MNIST dataset [57] from LIBSVM repository. Specifically, to convert the hand-written classification problem into a multi-task problem, we randomly chose 60000 images and divided them into $M$ agents. Each agent has two tasks, where each task has $n = 60000/(2*M)$ samples. Due to space limitations, we only present the convergence results for the case of non-i.i.d. data partition (i.e., data heterogeneity) and relegate the results of the i.i.d. data case to the appendix. For the non-i.i.d. data partition, we use the same data partition strategy as in [28], where each client can access data with at most two labels. In our experiments, a group of images is positioned in the top left corner, while another group of images is positioned in the bottom right. The two tasks are task "L" (to categorize the top-left digit) and task "R" (to classify the bottom-right digit). The overall problem is to classify the images of different tasks at different agents. All algorithms use the same randomly generated initial point. Here, we present experiments with $M = 10$ agents, where each agent has two tasks (i.e., $\mathbf{A} \in \mathbb{R}^{M\times 2}$ is an all-one matrix). We set the local update rounds $K = 10$. Experiments with a larger number of agents ($M = 5, 10, 30$) are provided here. The learning rates are chosen as $\eta_L = 0.1$ and $\eta_t = 0.1, \forall t$.

**2): River Flow Dataset and Learning Tasks:** We further test our algorithms on FMOL problems of larger sizes. In this experiment, we use the River Flow dataset[54], which is for flow prediction flow at eight locations within the Mississippi River network. Thus, there are *eight* tasks in this problem. In this experiment, we set $\eta_L = 0.001$, $\eta_t = 0.1$, $M = 10$, and keep the batch size $= 256$ while comparing $K$, and keep $K = 30$ while comparing the batch size. To better visualize 8 different tasks, we illustrate the normalized loss in radar charts in Fig. 2(b). We again verify that utilizing a larger training batch size and conducting additional local steps $K$ results in accelerated convergence.

**3): CelebA Dataset and Learning Tasks:** We utilize the CelebA dataset [55], consisting of 200K facial images annotated with 40 attributes. We approach each attribute as a binary classification task, resulting in a 40-way multi-task learning (MTL) problem. To create a shared representation function, we implement ResNet-18 [58] without the final layer, attaching a linear layer to each attribute for classification. In this experiment, we set $\eta_L = 0.0005$, $\eta_t = 0.1$, $M = 10$, and $K = 10$. Figure 3 displays a radar chart depicting the loss value of each binary classification task. In Figure 3, we demonstrate the efficacy of our FMGDA and FSMGDA algorithms in both i.i.d. case and non-i.i.d. case.

**Experiments on i.i.d. data:** First, we compare the convergence results with the same experimental settings in our Section. 5 but tested on the i.i.d data. As shown in Fig. 4, both FMGDA and FSMGDA

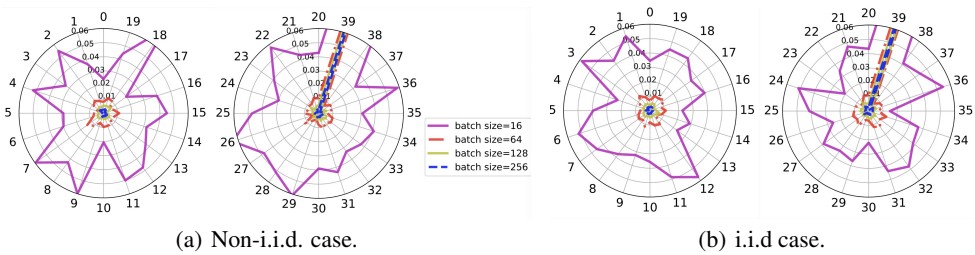

(a) Non-i.i.d. case.                            (b) i.i.d case.

Figure 3: Experiments on CelebA dataset.

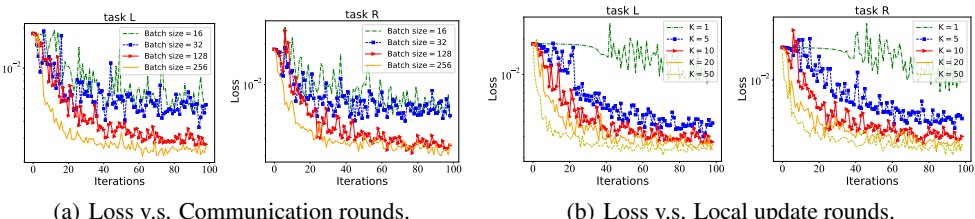

(a) Loss v.s. Communication rounds.         (b) Loss v.s. Local update rounds.

Figure 4: Experiments on i.i.d. data.

successfully converged in i.i.d. data, and the algorithm with a larger training batch size and more local updates $K$ may converge faster.

**Impact of the number of clients:** In this experiment, we choose the different number of clients from the discrete set $\{5, 10, 30\}$ and fix learning rates at $0.1$ and local update rounds at $10$. As shown in Fig. 5, a larger number of workers leads to faster convergence rates of our proposed algorithms both in i.i.d. case and non-i.i.d. case; this is mainly because more samples have been used while training while having more workers.

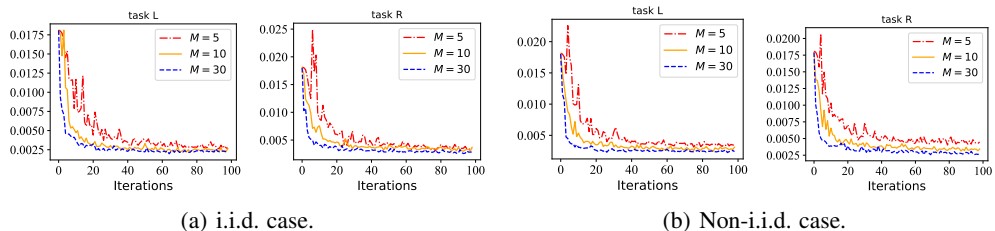

(a) i.i.d. case.                         (b) Non-i.i.d. case.

Figure 5: Loss value comparisons of algorithms on a different numbers of clients $M$.

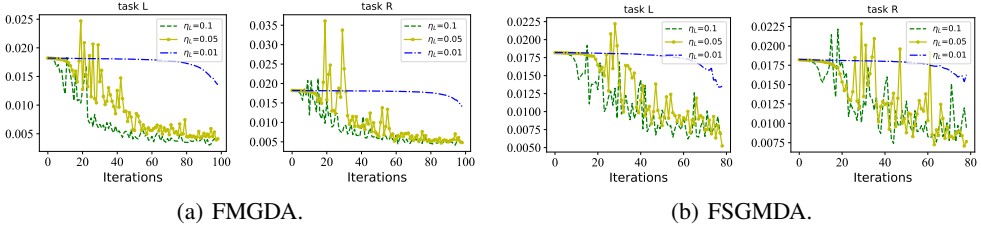

(a) FMGDA.                           (b) FSGMDA.

Figure 6: Comparisons of different step-sizes.

**Impact of the Step-size:** In this experiment, we choose the value of the learning rate $\eta_L$ from the discrete set $\{0.05, 0.01, 0.1\}$ and fix worker number at $5$, local update rounds at $10$. As shown in Fig. 6, larger local step-sizes lead to faster convergence rates on both FMGDA algorithm and FSMGDA algorithm.

