# Federated Multi-Objective Learning

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

662 $\qquad\qquad\qquad\qquad\qquad\qquad\qquad\qquad\qquad\qquad\qquad\qquad\qquad\qquad\qquad\qquad\qquad\qquad\qquad\square$

## 663 C.2 Non-convex Functions

664 **Theorem 5** (FSMGDA for Non-convex FMOL). *Let $\eta_t = \eta \leq \frac{3}{2(1+L)}$. Under Assumptions 4–6, if*
665 *an objective $f_s$ is bounded from below by $f_s^{\min}$, then the sequence $\{\mathbf{x}_t\}$ output by FSMGDA satisfies:*
666 $\min_{t \in [T]} \mathbb{E} \|\mathbf{d}_t\|^2 \leq \frac{2S(f_s^0 - f_s^{\min})}{\eta T} + \delta$, where $\delta = L\eta S^2 D^2 + S(\alpha \eta_L^2 K^2 D^2 + \beta \sigma^2)$.

667 *Proof.* Taking expectation on the random data samples conditioning on $\mathbf{x}_t$, we have

$$\mathbb{E} f_s(\mathbf{x}_{t+1}) \leq f_s(\mathbf{x}_t) + \mathbb{E} \left\langle \nabla f_s(\mathbf{x}_t), -\eta_t \sum_{j \in [S]} \lambda_j^t \Delta_j^t \right\rangle + \frac{1}{2} L \mathbb{E} \left\| \eta_t \sum_{j \in [S]} \lambda_j^t \Delta_j^t \right\|^2 \tag{87}$$

$$= f_s(\mathbf{x}_t) + \mathbb{E} \left\langle \nabla f_s(\mathbf{x}_t), -\eta_t \sum_{j \in [S]} \lambda_j^t \nabla f_j(\mathbf{x}_t, \xi_t) \right\rangle \tag{88}$$

$$+ \eta_t \mathbb{E} \left\langle \nabla f_s(\mathbf{x}_t), \sum_{j \in [S]} \lambda_j^t \left[ -\Delta_j^t + \nabla f_j(\mathbf{x}_t, \xi_t) \right] \right\rangle + \frac{1}{2} L \eta_t^2 \mathbb{E} \left\| \sum_{j \in [S]} \lambda_j^t \Delta_j^t \right\|^2 \tag{89}$$

$$\leq f_s(\mathbf{x}_t) - \eta_t \sum_{j \in [S]} \mathbb{E}[\lambda_j^t] \|\nabla f_j(\mathbf{x}_t)\|^2 \tag{90}$$

$$+ \eta_t \mathbb{E} \left\langle \nabla f_s(\mathbf{x}_t), \sum_{j \in [S]} \lambda_j^t \left[ -\Delta_j^t + \nabla f_j(\mathbf{x}_t, \xi_t) \right] \right\rangle + \frac{1}{2} L \eta_t^2 \mathbb{E} \left\| \sum_{j \in [S]} \lambda_j^t \Delta_j^t \right\|^2 \tag{91}$$

$$\leq f_s(\mathbf{x}_t) - \eta_t \sum_{j \in [S]} \mathbb{E}[\lambda_j^t] \|\nabla f_j(\mathbf{x}_t)\|^2 + \frac{\eta_t}{2} S \mathbb{E} \|\lambda_s^t \nabla f_s(\mathbf{x}_t)\|^2 \tag{92}$$

$$+ \frac{\eta_t}{2} \sum_{j \in [S]} \mathbb{E} \|\nabla f_j(\mathbf{x}_t, \xi_t) - \Delta_j^t\|^2 + \frac{1}{2} L \eta_t^2 \mathbb{E} \left\| \sum_{j \in [S]} \lambda_j^t \Delta_j^t \right\|^2 \tag{93}$$

$$\leq f_s(\mathbf{x}_t) - \frac{\eta_t}{2} \sum_{j \in [S]} \mathbb{E} \|\lambda_j^t \nabla f_j(\mathbf{x}_t)\|^2 + \frac{\eta_t}{2} \sum_{j \in [S]} \mathbb{E} \|\nabla f_j(\mathbf{x}_t, \xi_t) - \Delta_j^t\|^2 + \frac{1}{2} L \eta_t^2 \mathbb{E} \left\| \sum_{j \in [S]} \lambda_j^t \Delta_j^t \right\|^2. \tag{94}$$

668 Here we construct a virtual stochastic gradient $\nabla f_s(\mathbf{x}_t, \xi_t)$ with an independent sample. As
669 $\lambda_s^t$ only depends on $\Delta_s^t$, so $\lambda_s^t$ and $\nabla f_s(\mathbf{x}_t, \xi_t)$ are independent, from which the first in-
670 equality follows. The second inequality is due to $ab \leq \frac{1}{2}a^2 + \frac{1}{2}b^2$. Specifically,
671 $\mathbb{E} \left\langle \nabla f_s(\mathbf{x}_t), \sum_{j \in [S]} \lambda_j^t(-\Delta_j^t + \nabla f_j(\mathbf{x}_t, \xi_t)) \right\rangle = \sum_{j \in [S]} \mathbb{E} \left\langle \lambda_s^t \nabla f_s(\mathbf{x}_t), -\Delta_j^t + \nabla f_j(\mathbf{x}_t, \xi_t) \right\rangle \leq$
672 $\frac{S}{2} \mathbb{E} \|\lambda_s^t \nabla f_s(\mathbf{x}_t)\|^2 + \frac{1}{2} \sum_{s \in [S]} \mathbb{E} \|\nabla f_s(\mathbf{x}_t, \xi_t) - \Delta_s^t\|^2$. Also, following the fact that
673 $\lambda_s^t \in [0, 1]$, we have $\eta_t \sum_{s \in [S]} \mathbb{E}[\lambda_s^t] \|\nabla f_s(\mathbf{x}_t)\|^2 \geq \eta_t \sum_{s \in [S]} \mathbb{E}[(\lambda_s^t)^2] \|\nabla f_s(\mathbf{x}_t)\|^2 =$
674 $\eta_t \sum_{s \in [S]} \mathbb{E} \|\lambda_s^t \nabla f_s(\mathbf{x}_t)\|^2$. We also note that there exist a task $s$, such that $\frac{S}{2} \mathbb{E} \|\lambda_s^t \nabla f_s(\mathbf{x}_t)\|^2 \leq$
675 $\frac{1}{2} \sum_{j \in [S]} \mathbb{E} \|\lambda_j^t \nabla f_j(\mathbf{x}_t)\|^2$, which leads to the last inequality.

Rearranging the terms, we have