# OpenReview forum: "Federated Multi-Objective Learning"
_NeurIPS.cc/2023/Conference — NeurIPS 2023 poster_

### Official Review · Reviewer_nQxG · 2023-06-19

**Soundness:** 4 excellent
**Presentation:** 4 excellent
**Contribution:** 4 excellent
**Rating:** 7
**Confidence:** 3

**Summary:**

The paper proposes a new federated multi-objective learning (FMOL) framework where multiple clients can collaboratively solve an MOO problem while keeping their training data private. The authors propose two new federated multi-objective optimization (FMOO) algorithms called federated multi-gradient descent averaging (FMGDA) and federated stochastic multi-gradient descent averaging (FSMGDA) that allow local updates to reduce communication costs while achieving the same convergence rates as those of their algorithmic counterparts in the single-objective federated learning. The authors claim that their work is the first systematic attempt to bridge the gap between federated learning and MOO.

**Strengths:**

The paper addresses an important and timely problem of extending multi-objective optimization to the federated learning paradigm, considering both objective and data heterogeneity.

The paper proposes two new FMOO algorithms, FMGDA and FSMGDA, with provable Pareto-stationary convergence rate guarantees. Theoretical analysis of convergence rates looks reasonable.

The effectiveness of the algorithms is supported with comprehensive experimental results.

**Weaknesses:**

I find no big issues. Maybe assumption 4 is not so common.

**Questions:**

No.

**Limitations:**

Yes, limitations are addressed.

---

> ### Author Rebuttal · Authors · 2023-08-10
>
> Thank you very much for the positive review and the constructive comments.
>
> > I find no big issues. Maybe assumption 4 is not so common.
>
> **Our response:** Thanks for the positive review. The fact that our $(\alpha, \beta)$-Lipschitz continuous stochastic gradient assumption (Assumption 4) being uncommon is one of the novelties for our paper. Assumption 4 is an natural generalization of previous assumptions in MOO. The intuition behind Assumption 4 is that the rate of change of the stochastic gradient estimation stays within reasonable bounds and retains a satisfactory level of accuracy across any two independent samples. Thanks to this new assumption, we can provide the theoretical analyses in FMOO with data heterogeneity and objective heterogeneity.

---

> > ### Comment · Reviewer_kXdB · 2023-08-15
> >
> > Thanks for the clarification!

---

> > ### Comment · Reviewer_nQxG · 2023-08-16
> > **thanks**
> >
> > my evaluation remains the same

---

### Official Review · Reviewer_TvWz · 2023-06-28

**Soundness:** 3 good
**Presentation:** 3 good
**Contribution:** 2 fair
**Rating:** 4
**Confidence:** 3

**Summary:**

The authors  propose a federated multi-objective learning (FMOL) framework with multiple clients distributively and collaboratively solving an MOO problem while keeping their training data private. Notably, their FMOL framework allows a different set of objective functions across different clients to support a wide range of applications, which advances and generalizes the MOO formulation to the federated learning paradigm.

For this FMOL framework, they propose two new federated multi-objective optimization (FMOO) algorithms called federated multi-gradient descent averaging (FMGDA) and federated stochastic multi-gradient descent averaging (FSMGDA). Both algorithms allow local updates to significantly reduce communication costs, while achieving the same convergence rates as those of the their algorithmic counterparts in the single-objective federated learning.

**Strengths:**

 - The motivation is clear and this paper is well written.
 - The theoretical analysis of this paper is sufficient.

**Weaknesses:**

- How does the algorithm proposed in this paper solve the problem of data heterogeneity? In Algorithm 1, there is no design to solve data heterogeneity.
- Experimental results compared with FedMGDA+ [1] are lacking. FedMGDA+ is also suitable for the experimental setting in this paper.
- Experiments with different tasks for each client are missing. In the experiments of this work, each client has the same task. Since the authors claim that they are a generic algorithm that can deal with different setting, the experiments should be more comprehensive.




[1] Z. Hu, K. Shaloudegi, G. Zhang, and Y. Yu, “Federated learning meets multi-objective optimization,” IEEE Transactions on Network Science and Engineering, 2022.




**Questions:**

 - Line 52-54, what this challenge means? What means "FMOL algorithms could be extremely sensitive to small perturbations in the determination of common descent direction among all objectives"? Can the authors give a more detailed explanation?
 - Since each client may perform different tasks, how can the authors guarantee the network used in each client is the same?

**Limitations:**

See Weaknesses and Questions.

---

> ### Author Rebuttal · Authors · 2023-08-10
>
> Thank you very much for the review and the constructive comments. We believe the valuable suggestions from the reviewer have helped us significantly improve the quality of this paper. The detailed point-by-point responses are as follows:
>
> > 1. How does the algorithm proposed in this paper solve the problem of data heterogeneity? In Algorithm 1, there is no design to solve data heterogeneity.
>
> **Our response:** Thanks for your comment. In Algorithm 1, our basic idea and key technique to mitigate data heterogeneity is the use of a *two-sided learning rates*, where a local learning rate $\eta_L$ is used for local update (Step 2 in Algorithm 1) and a global learning rate $\eta_t$ is used for global update (Step 7 in Algorithm 1). With this two-sided learning rate approach, we can set a relatively small local learning rate $\eta_L$ to allow more local steps without local models drifting too far across clients. Also, we can set a relatively large global learning rate $\eta_t$ to ensure sufficiently fast convergence progress. By doing so, the "model drift" problem due to data heterogeneity could be properly controlled.
>
> > 2. Experimental results compared with FedMGDA+ [R1] are lacking. FedMGDA+ is also suitable for the experimental setting in this paper.
>
> **Our response:** Thanks for your comment. Our paper formulates a general federated multi-objective learning (FMOL) framework that is different from the setting in [R1]. Thus, our algorithms are *not* directly comparable to FedMGDA+. Specifically, we consider a FL system with $M$ clients and each client targets at $S$ tasks/objectives. In contrast, Ref. [R1] *reformulates* the classic single-objective FL as a multi-objective optimization problem, where each client has one distinctive objective. Under our experimental setting, it is unclear how to modify FedMGDA+ to allow multiple objectives at each client.
>
> [R1] Z. Hu, K. Shaloudegi, G. Zhang, and Y. Yu, “Federated learning meets multi-objective optimization,” IEEE Transactions on Network Science and Engineering, 2022.
>
> > 3. Experiments with different tasks for each client are missing. In the experiments of this work, each client has the same task. Since the authors claim that they are a generic algorithm that can deal with different setting, the experiments should be more comprehensive.
>
> **Our response:** Thanks for your comment. There appears to be some misunderstandings on our experiments. In our experiments, each client has the *same set of multiple tasks* rather a single same task. For example, each client has two tasks for the MultiMNIST dataset, eight tasks for the River dataset, and 40 tasks for the CelebA dataset. Also, these could be different across clients. Thus, our experiments already considered the *generic* settings consistent with our the FMOL formulation in Sec. 3.2.
>
> > 4. Line 52-54, what this challenge means? What means "FMOL algorithms could be extremely sensitive to small perturbations in the determination of common descent direction among all objectives"? Can the authors give a more detailed explanation?
>
> **Our response:** Thanks for your comment. One of the key challenges for gradient-based multi-objective optimization (MOO) is to compute an accurate common descent direction that can decrease all the objectives at the same time, given each objective's stochastic gradients. The common descent direction can be determined by solving a *quadratic programming* (cf. Step 7 in our Algorithm 1). However, the process of solving this quadratic programming to find a common descent direction is sensitive to each objective's stochastic gradient estimation. When such estimation is perturbed by noise, the common descent direction obtained from solving the quadratic programming could be biased and even lead to divergence issues. For example, even in centralized MOO, when the stochastic gradient has large Gaussian noise, divergence problems of many algorithms have been observed [5,13]. This situation is further excerbated by the data heterogeneity and objective heterogeneity as described in Line 45-52, which could inject additional bias in the evaluation of each objective's gradient estimation. As a result, FMOL algorithms could be extremely sensitive to perturbations and noises in the determination of common descent direction among all objectives.
>
> > 5. Since each client may perform different tasks, how can the authors guarantee the network used in each client is the same?
>
> **Our response:** Thanks for your comment. There appears to be some confusion between tasks (i.e., the $f(\cdot)$) and network (i.e., the $\mathbf{x}$-parameters) in our FMOL. To clarify, we start with the traditional multi-objective optimization (MOO). In MOO, the goal is to simultaneously optimize multiple objective functions (i.e., tasks) denoted as $\min_{x} (f_1(x), f_2(x), \cdots, f_S(x))$. That is, MOO seeks to find a *common solution* $\mathbf{x}$ (i.e., the network parameters) for the *entire set of objectives/tasks*. In other words, the network/model $\mathbf{x}$ is the same across all tasks. Following the same token, our FMOL extends the MOO problem setting to the federated learning paragdigm, where each client maintains a local copy of the same global network/model parameters (i.e., $\mathbf{x}$) for all tasks across the federated learning system. Thus, the network used in each client is the same *by design*.

---

> ### Author Response · Authors · 2023-08-20
>
> We'd like to thank the reviewer for the comments and feedbacks. Please let us know if you have any further new comments and questions. We are pleased to engage in a discussion regarding certain aspects of the work and provide clarification where needed.

---

> > ### Comment · Reviewer_TvWz · 2023-08-21
> >
> > Thanks for the authors' response. I'm still confused about this work in some points.
> >  - How the proposed two-sided learning rate approach can solve the data heterogeneity problem? Why a small learning rate for local update and a large learning rate for global update can solve the data heterogentity problem in FL? Furthermore, why a relatively small local learning rate can nake more local steps without local models drifting too far across clients? What means 'without local models drifting too far across clients'?
> > - For the advised compared related work [R1]. It's easy to extand FedMGDA to your setting that each client have multiple task. I still think it's better to compare and discuss with this related work since the proposed method is too similar to FedMGDA.
> >  - As the authors responsed that the network used in each client is the same by design. In this way, how different clients can solve different task? Is it necessary to ensure that different clients have the same output space? The ideal different tasks should means they have different output space. Thu authors are advised to consider this setting.

---

> > > ### Author Response · Authors · 2023-08-21
> > >
> > > Thanks for the reviewer's responses.
> > >
> > > > How the proposed two-sided learning rate approach can solve the data heterogeneity problem? Why a small learning rate for local update and a large learning rate for global update can solve the data heterogentity problem in FL? Furthermore, why a relatively small local learning rate can nake more local steps without local models drifting too far across clients? What means 'without local models drifting too far across clients'?
> > >
> > > **Our response:** Thanks for your comments. To fully understand why the two-sided learning rate approach can handle data heterogeneity, it is necessary to start with "model drift," which is a direct consequence of data heterogeneity under FL and leads to a performance degredation. Specifically, consider a client $i$ with a particular objective $f(\cdot)$ (we omit the objective index here for convenience) in communication roudn $t$. In standard FedAvg-type approaches, the evolution of the local model $x_{t,k}^i$ at client $i$ follows the trajectory $x_{t, k+1}^i = x_{t, k}^i - \eta_L g(x_{t, k}^i)$, where $g(x_{t, k}^i)$ represents the local stochastic gradient of the objective $f(\cdot)$. Due to data heterogeneity, the local stochastic gradient is no longer an unbiased estimate of the full gradient of the global loss function $f(\cdot)$ at $x_k$. Rather, the local stochastic gradient is only an unbiased estimator of the local full gradient, which could have significant variations across clients. As a result, the client $i$'s local model $x_{t, K}^i$ may progressively drift away from the global optimal solution and approach the local optimal solution that depends on the local data at client $i$, hence the name "model drift." The "model drift" phenomenon is the fundamental challenge under data heterogeneity in FL.
> > >
> > > The **basic idea** of the two-sided learning rate approach is to seperate the update schedules at local and global levels to avoid the model drift problem. Specifically, i) we can set a relatively **small** local learning rate $\eta_L$ so that the local models at each client do not drift too much toward their local optimal. Thanks to this smaller $\eta_L$, one can run more local steps without large model drifts while achieving good communication efficiency; and ii) On the other hand, one can set a relatively **large** global learning rate $\eta_t$ to ensure sufficient progress toward convergence of the global optimal solution. In summary, this two-sided learning rate approach reconciles the tension between a) running more local steps (better communication efficiency) and ii) suffering "model drift."

---

> > > > ### Author Response · Authors · 2023-08-21
> > > >
> > > > > For the advised compared related work [R1]. It's easy to extand FedMGDA to your setting that each client have multiple task. I still think it's better to compare and discuss with this related work since the proposed method is too similar to FedMGDA.
> > > >
> > > > **Our response:** Thanks for the new comment. Regarding FedMGDA+ in [R1], we highlight the following two major differences that prevent a straightforward comparison:
> > > >
> > > > 1. **Different Goals and Settings:** [R1] targets at optimizing *a single objective* among $m$ clients, while our work is to *simultaneously* $S$ objectives among $m$ clients. Specifically, [R1] reformulated the standard FL in the form of MOO so that a MGD-type can be used, i.e., [R1] considers $\min_{x} (f_1(x), f_2(x), \cdots, f_m(x))$, where $m$ is the number of clients. Here, the loss function $f_i(\cdot)$ at each client $i$, $i \in [m]$ are all corresponding to the **the same global task $f(\cdot)$** and the only difference stems from the heterogeneous data, i.e., $f_i(x) = f(x, D_i)$ where $D_i$ is the local data distribution that are different across $m$ clients. In contrast, our work is to solve $\min_{x} (f_1(x), f_2(x), \cdots, f_S(x))$, where the objectives $f_j(\cdot), j \in [S]$ are corresponding to **different tasks**, which are distributed to clients, i.e., $f_j(\cdot) = \frac{1}{m} \sum_{i \in [m]} f_{i, j}(\cdot)$, where $f_{i,j}(\cdot), i \in [m]$ corresponds to the *same task* as in $f_j(\cdot)$ but differ only in datasets from client to client.
> > > >
> > > >
> > > > 2. **Different Algorithmic Designs:** FedMGDA+ in [R1] is an *$\epsilon$-constraint approach* while our algorithm is not. Specifically, FedMGDA+ utilizes an important hyper-parameter $\epsilon$ (see Step 6 in FedMGDA+) to balance *different clients'* performance for the *same objective*, such as fairness, robustness and others (see discussions in their Sec.5). Our algorithm is to directly optimize $S$ objectives and achieve the Pareto-stationary solutions among different objectives. These algorithmic differences yield several problems to directly apply FedMGDA+ to our settings:
> > > >    * It is unclear how to set the $\epsilon$ hyper-parameter. For our setting, we have $S$ objectives and $m$ clients. If we directly apply FedMGDA+ to this setting, we need to have $S$ hyper-parameters $\epsilon_i, i \in [S]$, each of which targets at one objective $i \in [S]$ to balance the performance among clients $j \in [m]$.
> > > >    * Naively applying FedMGDA+ to our setting could generate a "two-level composite MOO": 1) we use FedMGDA+ to construct a common descent gradient to balance the performance among clients for each one specific objective, same as FedMGDA+ does in the original paper; and 2) we use MOO algorithms (e.g., FedMGDA+ or others) again to calculate a global descent direction based on results in 1) to solve the multi-objectives problem.  We believe this adaptation has already changed FedMGDA+ significantly, which no long can be called a fair comparison.
> > > >
> > > >    When we explore FedMGDA+ in our previous experiments, we can not make FedMGDA+ converge in our setting. Due to the time limitation in the rebuttal/discussion period, we can not carefully tune all parameters in all datasets. But here, we provide a simple example with the following setting: Consider two objectives $f_1(x) = (x-3)^2$ and $f_2(x) = (x+3)^2$ with two clients in the system. Each client could have an unbiased estimation of the gradient for each objective with Gaussian noise. With learning rate $0.01$, our algorithm could converge to the Pareto stationary point with $10^{-16}$ accuracy in less than 100 rounds, while FedMGDA+ **fails** to converge. This is not surprising because FedMGDA+ is not designed for this purpose. Thus, we believe it is not straightforward to compare FedMGDA+ with our algorithm in our setting.
> > > >
> > > > 3. **Relationship with [R1] under Special Settings:** We note we have compared with [R1] "qualitatively" in our paper. Specifically, in the related works, we mentioned [R1] and highlighted the different setting of [R1] in Lines 135 -138, i.e., "Although not directly related, classic FL has been reformulated in the form of MOO[30], which allows the use of a MGD-type algorithm nstead of vanilla local SGD to solve the standard FL problem." Also, when introducing our FMOL framework, we mentioned that [R1] is a special case of our setting in Line 188.

---

> > > > > ### Author Response · Authors · 2023-08-21
> > > > >
> > > > > > As the authors responsed that the network used in each client is the same by design. In this way, how different clients can solve different task? Is it necessary to ensure that different clients have the same output space? The ideal different tasks should means they have different output space. Thu authors are advised to consider this setting.
> > > > >
> > > > > **Our response:** In the definition of MOO (or our FMOL), the goal is to simultaneously optimize multiple objective functions (i.e., tasks) denoted as $\min_{x} (f_1(x), f_2(x), \cdots, f_S(x))$ where each objective $f_j: X \rightarrow C_j$ is distinct with the *same input space* $X$ (i.e., model parameters) but *different output space* $C_j$. That means, *different tasks can have different output spaces in MOO and our work by definition*. In other words, we do not have any assumptions or requirements on the output space for different tasks in our setting, and thus different clients can have totally different output spaces for different tasks.
> > > > >
> > > > > We also want to highlight that MOO is similar to many other *"multi-task learning (MTL) paradigms"*, where one trains a *single* model/network for *multiple* tasks. Specifically, when applying MOO in deep learning with neural networks, it is common to use the same neural network to solve different tasks simultaneously due to many reasons, such as parameter sharing, efficiency, among others. Take MultiMNIST datasets as an example, which is also detailed in our paper (Line 682-693). MultiMNIST dataset and its objectives are defined as follows: we randomly pick two images with different digits from the original MNIST dataset, and then combine these two images into a new one by putting one digit on the top-left corner and the other one on the bottom-right corner. Each digit can be moved up to 4 pixels on each direction. In this case, we have a two-objective problem to classify the item on the top-left (task 1) and to classify the item on the bottom-right (task 2). Typically, one neural network is used to sovle these two tasks simultaneously. Likewise, in our FMOL setting, the client can use one network to solve all different tasks.

---

### Official Review · Reviewer_gLj4 · 2023-07-04

**Soundness:** 3 good
**Presentation:** 3 good
**Contribution:** 2 fair
**Rating:** 3
**Confidence:** 3

**Summary:**

This paper generalizes multi-objective optimization to the federated learning paradigm, and proposes an algorithm for both full-batch and stochastic settings. Theoretical analysis shows same convergence rates as those of their algorithmic counterparts in the single-objective federated learning.

**Strengths:**

1. This paper studies an important and unsolved problem. The proposed setting is meaningful for many practical usages.

2. The idea to use $(\alpha,\beta)$-Lipshitzness to eschew incorrect assumption in SMGD is insightful. Though I am not very convinced, it provides a different way to consider this problem.

**Weaknesses:**

1. Since [1] has shown that SMGD will not converge to Pareto optimal for some stochastic cases due to the incorrect assumption on Lipschitz continuity of $\lambda$, I am not convinced how the $(\alpha,\beta)$-Lipshitzness can help the algorithm to converge, as the paper claims that Algorithm 1 can reduce to SMGD. Does Assumption 1 can lead to Lipschitz continuity of $\lambda$?

2. (Similar to point 1) The appendix provides further analysis for SMGD under assumption 4. After I checked the appendix in [1], I found an empirical test with Gaussian stochastic noise more aligned with practical cases (figure 5). The results show that SMGD will not converge to Pareto optimal. This conclusion is contradictory to the claim in this paper, so I am concerned about whether the assumption is still not natural or the proof is wrong. Given that I think Gaussian stochastic noise cases satisfy Assumption 4 (not carefully checked), I suspect some proof details are incorrect.

[1] On the Convergence of Stochastic Multi-Objective Gradient Manipulation and Beyond. NeurIPS 2022

**Questions:**

1. What is the direction bias in FSMGDA after having assumption 4? (Lemma 2 in [1])

[1] Fernando et al. Mitigating gradient bias in multi-objective learning: A provably convergent approach. ICLR 2023

---

> ### Author Rebuttal · Authors · 2023-08-10
>
> Thank you for the review and constructive comments. We believe the valuable suggestions have helped us significantly improve the quality of this paper. The detailed point-by-point responses are as follows:
>
> > 1. Since [R1] has shown that SMGD will not converge to Pareto optimal for some stochastic cases due to the incorrect assumption on Lipschitz continuity of $\lambda$, I am not convinced how the $(\alpha, \beta)$-Lipshitzness can help the algorithm to converge, as the paper claims that Algorithm 1 can reduce to SMGD. Does Assumption 1 can lead to Lipschitz continuity of $\lambda$?
>
> **Our response:** Thanks for your comments. There appears to be some misunderstandings on the assumptions used in this paper, and there is no contradiction with the claims in [R1]. Since the reviewer's questions are relatively complicated, we structure our responses in three parts as follows:
>
> 1. Our Assumption 1, $L$-Lipschitz continuous gradient assumption, is a standard assumption in optimization that characterizes the smoothness of a function. It is important to note that this assumption is completely different from the $\lambda$-Lipschitz continuity assumption, which focuses on the properties of $\lambda$, which is an implicit function of the stochastic gradient updates. Also, our $(\alpha, \beta)$-Lipschitz continuous stochastic gradient assumption (Assumption 4) is also completely different from the $\lambda$-Lipschitz continuity assumption (see below for further details). Thus, our Assumptions 1 and 4 do *not* imply or lead to the $\lambda$-Lipschitz continuity mentioned in [R1].
>
> 2. The $\lambda$-Lipschitz continuity is never used in our paper. Instead, we propose a new $(\alpha, \beta)$-Lipschitz continuous stochastic gradient assumption in Assumption 4. The intuition behind Assumption 4 is that the rate of change of the stochastic gradient estimation stays within reasonable bounds and retains a satisfactory level of accuracy across any two independent samples. Thanks to this new assumption, our proofs will not run into any incorrect use of the $\lambda$-Liptschitzness assumption, even when the our algorithm reduces to SMGD in special cases.
>
> 3. We note that the case with a large variance in stochastic gradients as shown in [R1] does not apply in this work. Specifically, our assumptions introduces a modest criterion for the stochastic gradient estimation that preclude cases with large variance in stochastic gradients. To see why this is the case and for a deeper understanding of our assumptions, please see our detailed discussions outlined in Lines 315-327, the systematic comparative analysis with other assumptions in Lines 556-599 in the Appendix, and our response to the reviewer's Comment 2.
>
> > 2. (Similar to point 1) The appendix provides further analysis for SMGD under assumption 4. After I checked the appendix in [R1], I found an empirical test with Gaussian stochastic noise more aligned with practical cases (figure 5). The results show that SMGD will not converge to Pareto optimal. This conclusion is contradictory to the claim in this paper, so I am concerned about whether the assumption is still not natural or the proof is wrong. Given that I think Gaussian stochastic noise cases satisfy Assumption 4 (not carefully checked), I suspect some proof details are incorrect.
>
> **Our response:** Thanks for your comments. Again, we remark that there is actually no contradiction with the claims in [R1], and there is no issue in our proof and anaysis. Note that [R1] showed that stochastic gradient manipulation techniques, such as SMGD [R2], PCGrad [R3], and CAGrad [R4], fail to converge when the given samples contain Gaussian stochastic noise. However, it is important to note that the example in [R1] does *not* disprove the theoretical results in [R2, R3, R4]. This is due to the simple logical fact that the example in [R1] does *not* satisfy the respective sufficient conditions in [R1, R2, R3] for their theoretical convergence results to hold.
>
> The situation is also the same when comparing our results with [R1], i.e., the example in [R1] does not satisfy the sufficient conditions in our theoretical results. Specifically, in order to ensure Pareto stationary convergence in non-convex functions, Corollary 6 requires a sufficient condition that $\beta = \mathcal{O}(\eta)$. Applying this sufficient condition in the experiment settings in [R1], we have $\beta = \mathcal{O}(0.006/\sqrt{T})$. However, the large variance setting in Gaussian stochastic noise for the example in [R1] leads to $\beta = \Omega(1)$, which clearly violates our sufficient condition. Hence, there is no logical contradiction between our work and [R1], and the example constructed in [R1] cannot be used to disprove our theoretical results.
>
> [R1] Zhou, Shiji, et al. "On the convergence of stochastic multi-objective gradient manipulation and beyond." Advances in Neural Information Processing Systems 35 (2022): 38103-38115.
>
> [R2] Sener, Ozan, and Vladlen Koltun. "Multi-task learning as multi-objective optimization." Advances in neural information processing systems 31 (2018).
>
> [R3] Yu, Tianhe, et al. "Gradient surgery for multi-task learning." Advances in Neural Information Processing Systems 33 (2020): 5824-5836.
>
> [R4] Liu, Bo, et al. "Conflict-averse gradient descent for multi-task learning." Advances in Neural Information Processing Systems 34 (2021): 18878-18890.
>
>
> > 3. What is the direction bias in FSMGDA after having assumption 4? (Lemma 2 in [R5])
>
> **Our response:** Thanks for your comment. The direction bias between the stochastic gradient and common descent gradient in FSMGDA is $H_{t, s} = \mathbb{E} \left\| \nabla f_s(x_t, \xi_t) - \Delta^t_s \right\|^2 \leq \alpha \eta_L^2 K^2 D^2 + \beta \sigma^2$ as shown in Lemma 3 (Line 647).
>
>
> [R5] Fernando, Heshan Devaka, et al. "Mitigating gradient bias in multi-objective learning: A provably convergent approach." The Eleventh International Conference on Learning Representations. 2022.

---

> ### Author Response · Authors · 2023-08-20
>
> We'd like to thank the reviewer for the comments and feedbacks. Please let us know if you have any further new comments and questions. We are pleased to engage in a discussion regarding certain aspects of the work and provide clarification where needed.

---

### Official Review · Reviewer_kXdB · 2023-07-05

**Soundness:** 3 good
**Presentation:** 4 excellent
**Contribution:** 3 good
**Rating:** 7
**Confidence:** 3

**Summary:**

The paper tackles the problem of Multi Objective Optimization in the Federated Learning Setting, which is a practically relevant problem. The proposed algorithms have same order of convergence rates to the single objective counterpart.

**Strengths:**

1. The motivation and related work is very well written.
2. A variety of experiment settings are considered which complements the theory very well.
3. It is interesting to see the convergence bounds are the same as the single objective counterpart.

**Weaknesses:**

There seem to be no major weaknesses of the paper. However, it would be good to highlight the non-trivial (if any) proof techniques needed to deduce the results. Also, refer to the questions.

**Questions:**

1. There is significant emphasis on client objective heterogenity, however the only occurence of this seems to be the range of the function f_s in the Theorem 1, which seems like very standard convergence rate analysis. Was there some investigation on how the constant factors in the rate deteriorate depending on the heterogenity of the client objectives, tham just the range of f_s?
2. Although the order of convergence is much better than the baselines presented in Table 1, I wonder why they were not compared against in the experiments? It would be a good check on the practicality of the constants in the bound in Theorem 1.

**Limitations:**

The work is technically solid and has no major limitations.

---

> ### Author Rebuttal · Authors · 2023-08-10
>
> Thank you very much for the review and the constructive comments. We believe the valuable suggestions from the reviewer have helped us significantly improve the quality of this paper. The detailed point-by-point responses are as follows:
>
> **General response about the constants in our results:** To assess the impact of the constants in our theorems, we conducted an ablation study encompassing various elements of the learning setting. This encompassed crucial parameters such as the number of local update steps ($K$), batch size, task number, data heterogeneity, number of clients, and learning rate. However, we note that certain constants tied to the objective functions, such as the Lipschitz constant, strongly-convex constant $\mu$, minimal function value $f_s^{min}$ and full gradient bound $G$, present challenges in quantification. More importantly, these constants are inherently tailored to the intricacies of each unique problem, rendering their interpretation less straightforward and less readily meaningful.
>
> > 1. There is significant emphasis on client objective heterogenity, however the only occurence of this seems to be the range of the function f_s in the Theorem 1, which seems like very standard convergence rate analysis. Was there some investigation on how the constant factors in the rate deteriorate depending on the heterogenity of the client objectives, than just the range of f_s?
>
> **Our response:** Thanks for your comments. The objective heterogeneity is not only about the function range of $f_s$, but also how each objective function $f_s$ is distributed accross clients by $f_{s, i}$ and its property with local dataset such as full gradient bound $G$. Due to these complications, it is hard to quantify the objective heterogeneity in a straightforward manner and then study its effect. To address this challenge, we provide ablation studies of all elements that could have direct impact on the objective heterogeneity, such as task number (different datasets), data heterogeneity (i.i.d. vs non-i.i.d. data), and number of clients.
>
> > 2. Although the order of convergence is much better than the baselines presented in Table 1, I wonder why they were not compared against in the experiments? It would be a good check on the practicality of the constants in the bound in Theorem 1.
>
> **Our response:** Thanks for your comments. In Table 1, the baselines were all designed for the centralized learning setting. First, we had an direct comparison with these baselines, as shown in Lines 379-386, Sec. 5 1-c. Moreover, it is worth noting that distributed version of these baselines can be interpreted as special cases of our FMGDA and FSMGDA algorithms with $K=1$ (no local update). So through the ablation studies of local udpate steps and data heterogeneity, the distributed version of the these baselines were also compared. See, e.g., Sec. 5 1-b, Table 2, and Fig 4(b) in the Appendix. According to these results, our algorithms have faster convergence than these baselines in different datasets.

---

> > ### Comment · Reviewer_kXdB · 2023-08-19
> >
> > Thanks for the response. All my concerns have been clarified.

---

### Official Review · Reviewer_8EfV · 2023-07-25

**Soundness:** 3 good
**Presentation:** 3 good
**Contribution:** 3 good
**Rating:** 6
**Confidence:** 3

**Summary:**

In this paper, the authors present a new distributed optimization problem named Federated Multi-Objective Optimization (FMOO) along with its corresponding algorithms. The idea behind FMOO is to reformulate the classical Multi-Objective Optimization (MOO) problem into a federated learning framework, where each client holds a specific subset of objectives. This formulation encompasses both existing centralized and federated MOO problems as special cases. In the proposed Multiple Gradient Descent Averaging strategy, each client calculates local updates for its respective objectives and transmits them to the server. The server, in turn, uniformly aggregates these local updates for each objective and subsequently constructs a combined set of updates that span across different objectives. This combination is achieved by solving a quadratic optimization problem.

The authors instantiate the strategy in the form of two algorithms: The Federated Multi-Gradient Descent Averaging (FMGDA) algorithm involves complete gradient evaluations within each client, and the Federated Stochastic Multi-Gradient Descent Averaging (FSMGDA) algorithm employs stochastic sampling. The authors provide a convergence analysis for both FMGDA and FSMGDA, encompassing scenarios with strongly-convex and non-convex objectives.


**Strengths:**

- Multi-objective optimization is a rapidly developing research area. The proposed approach extends the applicability and scope of federated learning, opening new possibilities for solving complex optimization tasks.
- The proposed algorithms are commendable for its simplicity and apparent straightforwardness of implementation.
- The authors provide theoretical convergence guarantees enhancing the significance of their work. The convergence analysis not only covers the proposed problem setting but also provides good insights into classical central MOO scenarios.
- The paper is generally well-written, although I have not thoroughly examined the proof. Including an appendix that addresses MOO challenges without heavily relying on external references is good.


**Weaknesses:**

- The communication cost per training interaction appears to be high for the proposed FMGDA algorithm, with a complexity of O(M x S) (M: number of clients, S: number of objectives). This cost could be a limiting factor, especially in scenarios with a large number of objectives and clients.
- Replicating the experiments in section 5 might be challenging based solely on the information provided in the main paper. Important details, including baseline learning algorithm specifics, are only available in the supplemental materials. Consolidating all important information in the main paper would improve accessibility.
- Minor comment: Table 3 might be better placed in the main paper. Additionally, it would be helpful to explicitly define $S_i$ before its usage (defined in the table).


**Questions:**

The current reviewer was wondering whether the parameter combination across objectives (as in Step 6 of the proposed algorithm) can be performed within each client, effectively reducing the communication cost per training iteration to O(M).

**Limitations:**

It would be beneficial if the authors added a paragraph explicitly discussing the limitations of the present study.

---

> ### Author Rebuttal · Authors · 2023-08-10
>
> Thank you very much for the review and the constructive comments. We believe the valuable suggestions from the reviewer have helped us significantly improve the quality of this paper. The detailed point-by-point responses are as follows:
> > 1. The communication cost per training interaction appears to be high for the proposed FMGDA algorithm, with a complexity of O(M x S) (M: number of clients, S: number of objectives). This cost could be a limiting factor, especially in scenarios with a large number of objectives and clients.
>
> **Our Response:** Thanks for your insightful comments. Communication cost is one of the major concerns in distributed learning systems. By reducing the communication frequency (i.e., utilizing local update steps), our FMOO algorithms can already reduce the total number of communication rounds by a factor of $K$, where $K$ is the local update steps.
>
> In terms of the communicaion cost per round, we agree with the reviewer that the per-round communication cost could be high with a large number of objectives and clients. However, such a high per-round communication cost is primarily due to the nature of the underlying FMOO learning system, rather than a limitation of our proposed algorithm for FMOO. In this paper, we focus on developing the first federated learning algorithm for FMOO with convergence guarantee. We will further consider communication-efficient FMOO in our future studies, which deserves an independent paper but beyond the scope of this work.
>
> > 2. Replicating the experiments in section 5 might be challenging based solely on the information provided in the main paper. Important details, including baseline learning algorithm specifics, are only available in the supplemental materials. Consolidating all important information in the main paper would improve accessibility.
>
> **Our Response:** Thanks for your comments. As you suggested, we will relocate the experimental description along with important details from the appendix to the main paper for a clear and accessible presentation in the revised version.
>
> > 3. Minor comment: Table 3 might be better placed in the main paper. Additionally, it would be helpful to explicitly define $S_i$ before its usage (defined in the table).
>
> **Our response:** Thanks for your suggestions. We will move Table 3 from the Appendix to the main paper and define $S_i$ before its first usage for a clear presentation of notations.
>
> > 4. The current reviewer was wondering whether the parameter combination across objectives (as in Step 6 of the proposed algorithm) can be performed within each client, effectively reducing the communication cost per training iteration to O(M).
>
> **Our response:** Thanks for your comments. We believe the problem of reducing the per-round communication cost to $O(M)$ is an interesting and promising topic for future exploration. However, we outline several challenges in pursuing this direction:
>
> 1. In our general FMOL framework, achieving $O(M)$ per-round communication cost is challenging. The challenges arise from the inherent *data heterogeneity* and *objective heterogeneity*, two characterize in FMOL problems. Specifically, due to data heterogeneity, each client $i$ can only possess an estimation of the local objective function $f_{s, i}$ corresponding to its local data, rather than the global function $f_s$ for each objective $s$. Consequently, this restricts the computation operation of parameter combinations to the set of objectives at each client. Specifically, each client can only obtain a shared descent direction for local objectives at each client ($f_{s, i}, \forall s \in [S]$, rather than $f_{s}, \forall s \in [S]$). Given the nonlinear nature of this process, the aggregation of these parameter combinations across multiple clients might need completely new approaches, obviously beyong simple averaging.
>
> 2. This challenge is compounded by situations in which distinct clients pursue varying sets of objectives, stemming from *objective heterogeneity*. That is, the parameter combinations across clients are the estimations of common descent directions for different objectives sets. In these cases, achieving a reduction in communication costs becomes even more challenging.
>
> While addressing the challenges above goes beyond the scope of our current paper, we appreciate the valuable insights from the reviewer, which could lead to new algorithms with low per-round communication cost different from traditional algorithm-agnostic approaches, such as gradient quantization and sparsification.
>
> > 5. It would be beneficial if the authors added a paragraph explicitly discussing the limitations of the present study.
>
> **Our response:** Thanks for the suggestion. We will add one paragraph to discuss the limitations in the revision.

---

> > ### Comment · Reviewer_8EfV · 2023-08-14
> >
> > I thank the authors for their detailed response. The majority of my concerns have been satisfactorily addressed through their response.
> > I still maintain reservations regarding the potential for a substantial communication overhead.
> > However, I believe that the initial approach towards formulating the multi-objective optimization (MOO) problem within a federated learning framework holds significant value.
> > To enhance the clarity of the contribution, I would suggest a more explicit comparison with [30], as indicated by the authors in their response to Reviewer TvWz.

---

> > > ### Author Response · Authors · 2023-08-15
> > >
> > > We appreciate the reviewer's recognition of the significance and originality of our federated multi-objective learning (FMOL) framework. For the communication overhead challenge, we find the reviewer's insights to be enlightening. These insights have the potential to pave the way for developing innovative communication-efficient algorithms. This would be a promising and interesting future work, which will differ from the conventional algorithm-agnostic gradient compression techniques.
> > >
> > > Regarding Ref. [30], we note that we have compared with Ref. [30] in our paper (see Lines 135 -138). Simply speaking, Ref. [30] reformulated the standard federated learning (FL) in the form of multi-objective optimization (MOO), which allows the use of a MGD-type algorithm instead of vanilla local SGD to solve the standard FL problem. Thus, the goal of Ref. [30] is quite different from that of our work, which is aiming to solve the more complex MOO in an FL setting, not the conventional FL problem. Moreover, we note that Ref. [30] can be covered as a *special case* under our proposed FMOL framework (see Line 188 in the paper). We appreciate the reviewer's suggestion and will include more explicit comparison with Ref. [30] in the revision to enhance the clearity of our contributions.

---

### Decision · Program_Chairs · 2023-09-21

**Decision:**

Accept (poster)

**Comment:**

This paper presents a formulation of a new problem, federated multi-objective learning, and proposes a method to solve the problem for the first time. This is a borderline case because the average score is close to the threshold and the review scores are significantly diverging, from 3 to 7. However, I believe that the formulation of a new problem can be regarded as an important contribution of this work although the formulation is kind of an extension to federated learning and the problem setting is not very realistic. The methodology to solve the problem is reasonable and shows its effectiveness, but it is also true that it has known limitations such as large communication costs. After rebuttal, a large portion of concerns appear to be addressed but there exists a remaining concern on the theoretical side. Considering all these facts, I would recommend accepting this paper to motivate the research community in federated learning towards this new problem besides acknowledging the technical contributions of this work.